# Diversity trapped in cages: Revision of *Blumenavia* Möller (Clathraceae, Basidiomycota) reveals three hidden species

**Gislaine C. S. Melanda**[1]*, **Thiago Accioly**[1], **Renato J. Ferreira**[2], **Ana C. M. Rodrigues**[2], **Tiara S. Cabral**[3], **Gilberto Coelho**[4], **Marcelo A. Sulzbacher**[4], **Vagner G. Cortez**[5], **Tine Grebenc**[6], **María P. Martín**[7], **Iuri G. Baseia**[1,2]

**1** Departamento de Botânica e Zoologia, Programa de Pós-Graduação em Sistemática e Evolução, Universidade Federal do Rio Grande do Norte, Natal, Rio Grande do Norte, Brazil, **2** Departamento de Micologia, Programa de Pós-Graduação em Biologia de Fungos, Universidade Federal de Pernambuco, Recife, Pernambuco, Brazil, **3** Programa de Pós-Graduação em Genética, Conservação e Biologia Evolutiva, Instituto Nacional de Pesquisas da Amazônia, Manaus, Amazonas, Brazil, **4** Universidade Federal de Santa Maria, Santa Maria, Rio Grande do Sul, Brazil, **5** Departamento de Biodiversidade, Universidade Federal do Paraná—Setor Palotina, Palotina, Paraná, Brazil, **6** Slovenian Forestry Institute, Ljubljana, Slovenia, **7** Departamento de Micología, Real Jardín Botánico-CSIC, Madrid, Spain

* gsmelanda@gmail.com

**Data Availability Statement:** All relevant data are within the manuscript.

**Funding:** This work was supported by: Coordenação de Aperfeiçoamento de Pessoal de

## Abstract

Basidiomata of Phallales have a diversified morphology with adhesive gleba that exudes an odor, usually unpleasant that attracts mainly insects, which disperse the basidiospores. The genus *Blumenavia* belongs to the family Clathraceae and, based on morphological features, only two species are currently recognized: *B. rhacodes* and *B. angolensis*. However, the morphological characters adopted in species delimitations within this genus are inconsistent, and molecular data are scarce. The present study aimed to review and identify informative characters that contribute to the delimitation of *Blumenavia* species. Exsiccates from America and Africa were analyzed morphologically, and molecularly, using ITS, LSU, *ATP*6, *RPB*2 and *TEF*-1α markers for Maximum Parsimony, Bayesian and Maximum likelihood analyses, and also for coalescent based species delimitations (BP&P), as well as for bPTP, PhyloMap, Topo-phylogenetic and Geophylogenetic reconstructions. According to our studies, seven species can be considered in the genus: *B. rhacodes* and *B. angolensis* are maintained, *B. usambarensis* and *B. toribiotalpaensis* are reassessed, and three new species are proposed, *B. baturitensis* Melanda, M.P. Martín & Baseia, sp. nov., *B. crucis-hellenicae* G. Coelho, Sulzbacher, Grebenc & Cortez, sp. nov., and *B. heroica* Melanda, Baseia & M.P. Martín, sp. nov. *Blumenavia rhacodes* is typified by selecting a lectotype and an epitype. Macromorphological characters considered informative to segregate and delimit the species through integrative taxonomy include length of the basidiomata, color, width and presence of grooves on each arm as well as the glebifer position and shape. These must be clearly observed while the basidiomata are still fresh. Since most materials are usually analyzed after dehydration and deposit in collections, field techniques and protocols to describe fugacious characters from fresh specimen are demanded, as well as the use of molecular analysis, in order to better assess recognition and delimitation of species in *Blumenavia*.

Nível Superior (CAPES) for ACMR, GCSM, TA and TSC scholarships, Conselho Nacional de Desenvolvimento Científico e Tecnológico (CNPq) for RJF scholarship; Special Visiting Researcher - PVE/CNPq (Process 407474/2013–7) given to IGB and MPM; Plan Nacional I+D+i project CGL2015-67459-P given to MPM; Brazil–Slovenia bilateral project (BI-BR/11-13-005(SRA)/490648/2010-0 (CNPq) and the Research Program in Forest Biology, Ecology and Technology (P4-0107) of the Slovenian Research Agency support TG and MS work.

**Competing interests:** The authors have declared that no competing interests exist.

## Introduction

The Phallales E. Fisch. are gasteroid fungi that commonly possess zoochoric dispersion of their basidiospores, through insects that are attracted by the unpleasant odor of the mucilaginous gleba. The usual appearance and smell gave rise to the popular denomination of "stinkhorns " or "cage fungi" [1, 2]. *Blumenavia* Möller is a genus belonging to this order and is included in the family Clathraceae Chevall. Möller [3] proposed this genus based on specimens discovered in Blumenau, Santa Catarina (Brazil), with *Blumenavia rhacodes* Möller as the type species; this author segregated *Blumenavia* from *Laternea* Turpin based on the pale-yellow basidiomata color, triangular to irregular wing-like glebifers occupying the external borders of the whole length of the arms, and a wide groove at the external face of the arms present in *B. rhacodes*.

Until now, based on morphological data, three other species have been recognized in the genus: *Blumenavia angolensis* (Welw. & Curr.) Dring [4], basionym *Laternea angolensis* Welw. & Curr. [5]; *B. usambarensis* Henn. [6]; and *B. toribiotalpaensis* Vargas-Rodr [7]. In addition, Dring [4] proposed *B. usambarensis* as a synonym of *B. angolensis*, since both present white basidiomata, absence of a groove in the outer face of the arms, and glebifers restricted to a quarter or a third of the receptacle; characters that differentiate them from *B rhacodes*. This taxonomic treatment proposed by Dring [4] has been widely adopted by subsequent studies [8–11]. However, despite the proposed synonymy, the author did not analyze the type specimen of *B. usambarensis*, from Tanzania, but only exsiccates from Brazil and Trinidad and Tobago.

The most recent species described, using as diagnostic characters color of basidiomata, size of basidiospores, arrangement of the gleba in the arms of the receptacle, and cross-section of the arm, was *B. toribiotalpaensis* [7]. However, Calonge et al. [12] synonymized *B. toribiotalpaensis* with *B. rhacodes*, considering as similar characters: basidiomata staining, number and anatomy of arms and basidiospore dimensions. This synonymy has been mostly adopted by later studies [10–13].

The scarcity of morphological characters adopted to segregate the species and the ephemeral basidiomata are limitations in studies on Phallales. The hypothesis of our work is that the use of morphological and molecular data, as well as geographical distribution, are effective to delimit species in the genus *Blumenavia*. In this context, the aim of the present study was to review collections of the genus *Blumenavia*, including type specimens and specimens from type localities, using morphological and molecular analyses, to revise the taxonomy and systematics of the genus.

## Materials and methods

### Sampling

For this study 23 exsiccates from Brazil (South and Northeast), Mexico (Jalisco and Xalapa), and Tanzania were analyzed, including type specimens, this collections were borrowed from: Botanischer Garten und Botanisches Museum Berlin (B); U.S. National Fungus Collections, USDA-ARS (BPI); Instituto de Botánica Universidad de Guadalajara (IBUG); Herbário ICN, Departamento de Botânica Universidade Federal do Rio Grande do Sul (ICN); Instituto de Ecología, A.C (XAL); Museu Botânico Municipal (MBM); Instituto Anchietano de Pesquisas/ UNISINOS (PACA) and Coleção de Fungos da Universidade Federal do Rio Grande do Norte (UFRN-Fungos).

Three materials were collected following methodology of previous articles [9,10]: UFRN--Fungos 2886 under the authorization SISBIO 55685; SMDB 18116 as well as ICN 200594

under the Centro de Pesquisas e Conservação da Natureza Pró-Mata's authorization. These material were depositad in UFRN-Fungos, SMDB (Herbário SMDB, Jardim Botânico da Universidade Federal de Santa Maria, Centro de Ciências Naturais e Exatas) and ICN, respectively.

All of those materials cited were included in the morphological analyses, and 13 were used in molecular analyses. All herbaria gave permissions for morphological and molecular studies.

## Morphological studies

Specific literature was consulted to determine the analyzed characters [3–21]. The colors of the structures follow Küppers [22]. For each specimen the size, color and texture of expanded basidiomata, receptacles, glebifers, volva, as well as rhizomorphs were analyzed at a macroscopic level, therefore are written in the description based on fresh basidiomata. In addition, microscopic observations were obtained from the hyphae forming the rhizomorphs, basal and apical arms, glebifers, basal and apical exoperidia, mesoperidia, and endoperidia. Since these characters were not informative to segregate species, microscopic analyzes of basal and apical arms, glebifers, basal exoperidia, mesoperidia and endoperidia are not included in taxonomical descriptions (see S1 Table). However, this study is the first in which the different peridium layers have been analyzed in *Blumenavia* (see layers detail in Fig 9A). In this paper, only the informative characters are included in the description of each species. These microstructures were mounted in 5% KOH (potassium hydroxide) and 1% Congo red. The shape and size of basidiospores of each specimen were observed mounted in 5% KOH, Melzer's reagent, cotton blue and 1% Congo red. These dyes can provide essential information through chemical reactions characteristic of certain taxa [1], also, it is the first time that microstructure reactions were analyzed in *Blumenavia*.

To describe the shape of basidiospores, Q value (height/width quotient) was calculated according to Bas [23]; Qm represents the mean Q value, where "*x*" is the mean of values previously presented, and "±" is the standard deviation, as observed under the 100× objective. For each microscopic structure thirty hyphae/basidiospores were measured for each basidiomata.

## Molecular studies

**DNA extraction, PCR amplification and sequencing.** DNA was isolated using Qiagen DNeasy® Plant Mini Kit (Germantown, MD, USA), following the manufacturer's instructions except lysis buffer incubation, which was made overnight at 55˚C. PuReTaq Ready-To-Go™ PCR Beads (GE Healthcare Life Sciences, NJ, USA) were used for PCR (Polymerase Chain Reaction) following Martín and Winka [24]. Internal Transcribed Spacer (ITS1-5.8S-ITS2 nrDNA; fungi barcode), nuclear ribosomal Large Subunit (LSU nrDNA), mitochondrial ATPase subunit 6 (*ATP*6), the gene region for the second largest subunit of RNA polymerase (*RPB*2), and translation elongation factor 1 alpha (*TEF*-1α) were amplified with the primers ITS5/ITS4 [25], LR0R/LR5 [26], ATP6-3/ATP6-4 [27], bRPB2-6F/bRPB2-7.1R [28], EF1-1018F/EF1-1620 [29], respectively.

ExoSAP-IT™ PCR Product Cleanup Reagent (USB Corporation, OH, USA) was used for purification of the amplifications, following the fabricant instructions. Sequencing of the PCR products was carried out at Macrogen in Seoul, South Korea. All the sequences are deposited in Genbank® (http://www.ncbi.nlm.nih.gov/).

**Phylogenetic analyses.** The sequences obtained in this study are listed in Table 1. These sequences were aligned in MEGA X through the Muscle algorithm followed by manual adjustment with the sequence of *B. angolensis* obtained from NCBI/GenBank database and published in Degreef et al. [8], and two *Clathrus* species included as outgroup. Then this dataset

was submitted to the Partition Homogeneity (Incongruence Length Difference) Test in software PAUP* v.4.0a166 [30].

Phylogenetic analyses were conducted for the concatenated five loci dataset under maximum parsimony (MP) analysis, Bayesian, and maximum likelihood (ML). Alignment gaps were marked with "–" and missing data were indicated with "N". The parameters used in the phylogenetic analyses were modified from Sousa et al. [31]. MP analysis was conducted in software PAUP*, initial trees for 10,000 bootstrap simulations (MPbs) of Heuristic Search under TBR swapping algorithm were obtained by 10x randomized stepwise addition, and the number of trees obtained in each bootstrap replication was not restrained. In the Bayesian analysis, performed using Mr.Bayes v.2.3.6 [32] implemented in CIPRES Science Gateway (Cyberinfrastructure for Phylogenetic Research web portal, http://www.phylo.org), partitions were set to use specific substitution models chosen by the software jModelTest 2 [33] for each marker under the Akaike Informational Criterion (AIC) and corrected Akaike Informational Criterion (AICc). Substitution rates, proportions of invariant sites, gamma shapes and base frequencies were independently estimated among partitions from initial priors obtained by jModelTest. Ten million generations were used, and trees were sampled each 5,000 generations. Convergence and sampling were checked by Potential Scale Reduction Factor (PSRF) and Estimated Sample Size (ESS) as recommended by MrBayes developers. Convergence was also visualized in Tracer v.1.7.1 [34]. Stationarity of MCMC were assessed in Tracer v.1.7.1 software and also manually in MCMC output from MrBayes v.2.3.6 when Average Standard Deviation of Split Frequencies (AvgStdDev) dropped below 0.01. In the Bayesian analysis we tested two partition models: one considering five partitions (ITS1-5.8S-ITS2/LSU/ATP6/RPB2/TEF-1a), and another considering seven partitions (ITS1/5.8S/ITS2/LSU/ATP6/RPB2/TEF-1a). ML analyses were performed using RAxML software [35] implemented in CIPRES Science Gateway (Cyberinfrastructure for Phylogenetic Research web portal, http://www.phylo.org) and 1,000 bootstrap replicates (MLbs) under GTRGAMMA model.

For poorly supported terminal clades in which morphological variation among terminals was inconspicuous, additional species delimitation analyses were carried out based on a reduced alignment without outgroups, to increase homology in alignment, and refine results in subtree analyses. The software BP&P v.3.3 (Bayesian Phylogenetic & Phytogeography version 3.3) [36] was used for simultaneous species recognition and delimitation by Bayesian Markov chain Monte Carlo (MCMC) under a multispecies coalescent model (MSC) using algorithms 0 and 1, and considering equal probabilities for the number of species and for the rooted species trees [37–39] by 100,000 replicates sampled every 5 generations and burnin of 2,000 sampled generations. *Theta* and *Tau* priors where adjusted after previous run of the program for estimation of parameters under the MSC model. MCMC step lengths where set to automatic adjustment during the burnin phase. Additionally, Bayesian analysis was carried out following the same methodology as in the general analysis to generate an input tree for bPTP (Bayesian Poisson Tree Processes) [40], PhyloMap and Topo-phylogenetic [41] reconstructions. Although bPTP was designed for single locus analysis, it is commonly used for multilocus trees [42–44]. To illustrate the distribution of *Blumenavia* spp. in the world, the Geophylogeny was made using the world map with the phylogenetic tree with CorelDraw® Graphics Suite X8 software.

The software FigTree (http://tree.bio.ed.ac.uk/software/figtree/) was used to view and edit the phylogenetic trees. Final editing of all images was done in CorelDraw® Graphics Suite X8 software. The final alignment and analysis were deposited in Treebase under ID 25096 (http://purl.org/phylo/treebase/phylows/study/TB2:S25936; for reviewer access: http://purl.org/phylo/treebase/phylows/study/TB2:S25936?x-access-code=a21d01f584a54a1997c0c42350b6f8f1&format=html).

## Nomenclature

The electronic version of this article in Portable Document Format (PDF) in a work with an ISSN or ISBN will represent a published work according to the International Code of Nomenclature for algae, fungi, and plants, and hence the new names contained in the electronic publication of a PLOS ONE article are effectively published under that Code from the electronic edition alone, so there is no longer any need to provide printed copies.

In addition, new names contained in this work have been submitted to MycoBank, from where they will be made available to the Global Names Index. The unique MycoBank number can be resolved and the associated information viewed through any standard web browser by appending the MycoBank number contained in this publication to the prefix at http://www.mycobank.org/MB. The online version of this work is archived and available from the following digital repositories: PubMed Central, LOCKSS and Digital-CSIC.

## Results and discussion

### Molecular phylogeny

For this study, DNA sequences of 13 specimens of *Blumenavia* were obtained: 12 ITS, 9 LSU, 7 *ATP*6, 10 *RPB*2 and 11 *TEF*-1a (Table 1), except *B. usambarensis*. Incongruence Length Difference test has a P value = 1.00. The concatenated alignment obtained has 3,803 unambiguously

**Table 1.** ***Blumenavia* specimens analyzed in molecular analysis.** Genbank accession numbers for each region of the sequences included in this study. Accession numbers in bold are from newly generated sequences. Absences of sequences are marked with "–".

| Taxa | Voucher | Country, State | ITS | LSU | *ATP6* | *RPB2* | *TEF-1α* |
|---|---|---|---|---|---|---|---|
| *Blumenavia angolensis* | BR JD772 | São Tomé and Principe, São Tomé | - | KC128653 | - | - | - |
| *Blumenavia baturitensis* sp. nov. | UFRN-Fungos 1943-paratype | Brazil, Ceará | **MG817726** | **MG817734** | **MH061925** | **MH061934** | **MH061943** |
| | UFRN-Fungos 2868-holotype | Brazil, Ceará | **MG817725** | **MG817733** | **MH061924** | **MH061933** | **MH061942** |
| *Blumenavia crucis-hellenicae* sp. nov. | SMDB 18116—holotype | Brazil, Rio Grande do Sul | **LN875253** | **MK958819** | **MK975457** | **MK975458** | **LN875257** |
| | ICN 200594—paratype | Brazil, Rio Grande do Sul | **LN875254** | - | - | - | **LN875258** |
| | ICN 177268—paratype | Brazil, Rio Grande do Sul | - | **MG817727** | **MH061920** | **MH061926** | **MH061935** |
| | ICN 177269—paratype | Brazil, Rio Grande do Sul | **MG817717** | **MG817728** | **MH061921** | **MH061927** | - |
| *Blumenavia heroica* sp. nov. | XAL S. Chacón 5257-A—paratype | Mexico, Veracruz | **MG817720** | **MG817731** | - | **MH061929** | **MH061938** |
| | XAL E. Gándara 1455—holotype | Mexico, Veracruz | **MG817721** | - | - | **MH061930** | **MH061939** |
| | XAL D. Jarvio 778—paratype | Mexico, Veracruz | **MG817722** | - | - | - | - |
| *Blumenavia rhacodes* | ICN 176968 | Brazil, Rio Grande do Sul | **MG817718** | **MG817729** | **MH061922** | - | **MH061936** |
| | ICN 177266—epitype | Brazil, Rio Grande do Sul | **MG817719** | **MG817730** | **MH061923** | **MH061928** | **MH061937** |
| *Blumenavia toribiotalpaensis* | BPI 870955—holotype | Mexico, Jalisco | **MG817724** | **MG817732** | - | **MH061932** | **MH061941** |
| | IBUG 456—paratype | Mexico, Jalisco | **MG817723** | - | - | **MH061931** | **MH061940** |
| Outgroups (*Clathrus archeri* and *Clathrus chrysomycelinus*) | *C. archeri* KOR C_A005 | Poland, Marwice | KP688380 | KP688386 | - | - | - |
| | *C. chrysomycelinus* PDD 75096 | New Zealand | - | DQ218626 | DQ218915 | DQ219083 | DQ219262 |

aligned nucleotide positions, of which 3,573 were constant, 139 were parsimony uninforma-tive, and 91 were parsimony informative. Maximum Parsimony analysis resulted in a most parsimonious tree (not shown) with 250 steps (CI = 0.948, RI = 0.933, RC = 0.884, HI = 0.052). Maximum likelihood analysis gave a best tree with -lnL = -6,640.394 (not shown). MP and ML trees can be viewed in TreBase: http://purl.org/phylo/treebase/phylows/study/TB2:S25936; for reviewer access: http://purl.org/phylo/treebase/phylows/study/TB2:S25936?x-access-code= a21d01f584a54a1997c0c42350b6f8f1&format=html. Bayesian analyses considering two parti-tion models retrieved the same topology. The posterior probabilities result was slightly higher in the analysis with seven partition, so we chose to show only the results for this analysis (Fig 1). Bayesian analysis gave a best tree with -lnL = -6,633.82. The supports of the other two anal-ysis as also show in the Fig 1.

In the three phylogenetic of all analyses (MP, ML, BI), five clades and a singleton were recovered for the genus *Blumenavia* (Fig 1), corresponding to the three known species (*B. angolensis*, *B. rhacodes*, *B. toribiotalpaensis*), and three species that we consider new to science: *B. baturitensis* sp. nov., *B. crucis-hellenicae* sp. nov., and *B. heroica* sp. nov. Also, the morpho-logical analysis made it possible to confirm *B. usambarensis*, as an independent species; it was not possible to obtain molecular data from these old specimens, which were collected in 1900.

MP, ML and BI show *B. baturitensis* sp. nov., *B. heroica* sp. nov. and *B. rhacodes* grouped together in a clade, but in ML *B. angolensis* is grouped in the middle with *B. rhacodes* with low support. In MP *Blumenavia angolensis* forms a singleton in a bigger clade of *B. baturitensis*, *B. rhacodes* and *B. heroica*, and in BI as a singleton together with the clades *B. toribiotalpaensis* and *B. crucis-hellenicae*. An explanation is that *B. angolensis* has only LSU sequence, in contrast with more than one with the others.

*Blumenavia toribiotalpaensis* and *B. crucis-hellenicae* are grouped in a clade in BI and ML, in BI this clade shares a common ancestor with *B. angolensis*, as mentioned before. In MP *B. crucis-hellenicae* is a sister group of the clade with *B. angolensis* and the one with *B. baturiten-sis*, *B. rhacodes* and *B. heroica*, *B. toribiotalpaensis* in this analysis appears out of all these clades.

An evolutionary relationship for *B. angolensis* may be unclear because the lack of some markers, but these differences between the trees don't reveal an incongruency in the separation of the species. Defining infrageneric relationship was not intended in this work, so evolution-ary placement of *B. angolensis* can made in future works with better sampling of this species.

As shown in Fig 1, *Blumenavia baturitensis* sp. nov., *B. heroica* sp. nov. and *B. rhacodes* grouped together in a clade, within three independent groups, but with low support values; and, since these species show few unremarkable morphological differences, in order to test species delimitation hypothesis, this clade was chosen for secondary analyses. The BP&P analy-sis obtained a posterior probability of 0.9746 for three species in the dataset, which is visualized in Fig 2 with bPTP Phylomap program; the same results were obtained (three species recorder) using Topo-Phylogeny.

Additional data that supports the three species delimitation, as well as in the case of the other species in *Blumenavia*, are regarding the geographical distribution (Fig 3). *Blumenavia angolensis* identified in the Southern and Northeastern Brazil are the new species *B. crucis-hel-lenicae* sp. nov. and *B. baturitensis* sp. nov., respectively. Both segrageted from *B. angolensis* from Africa. The type location of *Blumenavia rhacodes* is Southern Brazil, collection from this region maintains its identity and was shown to be distinct from others identified as *B. rhacodes* from Mexico; the latter is a novelty for science: *B. heroica* sp. nov. In addition, in Mexico, *B. toribiotalpaensis* was considered by other authors [10–13] as *B. rhacodes*, but both were shown to be distinct, assuming the invalidation of this synonymization. Given the restrict distribution found for all species in our revision on genus *Blumenavia*, our results rise the possibility for

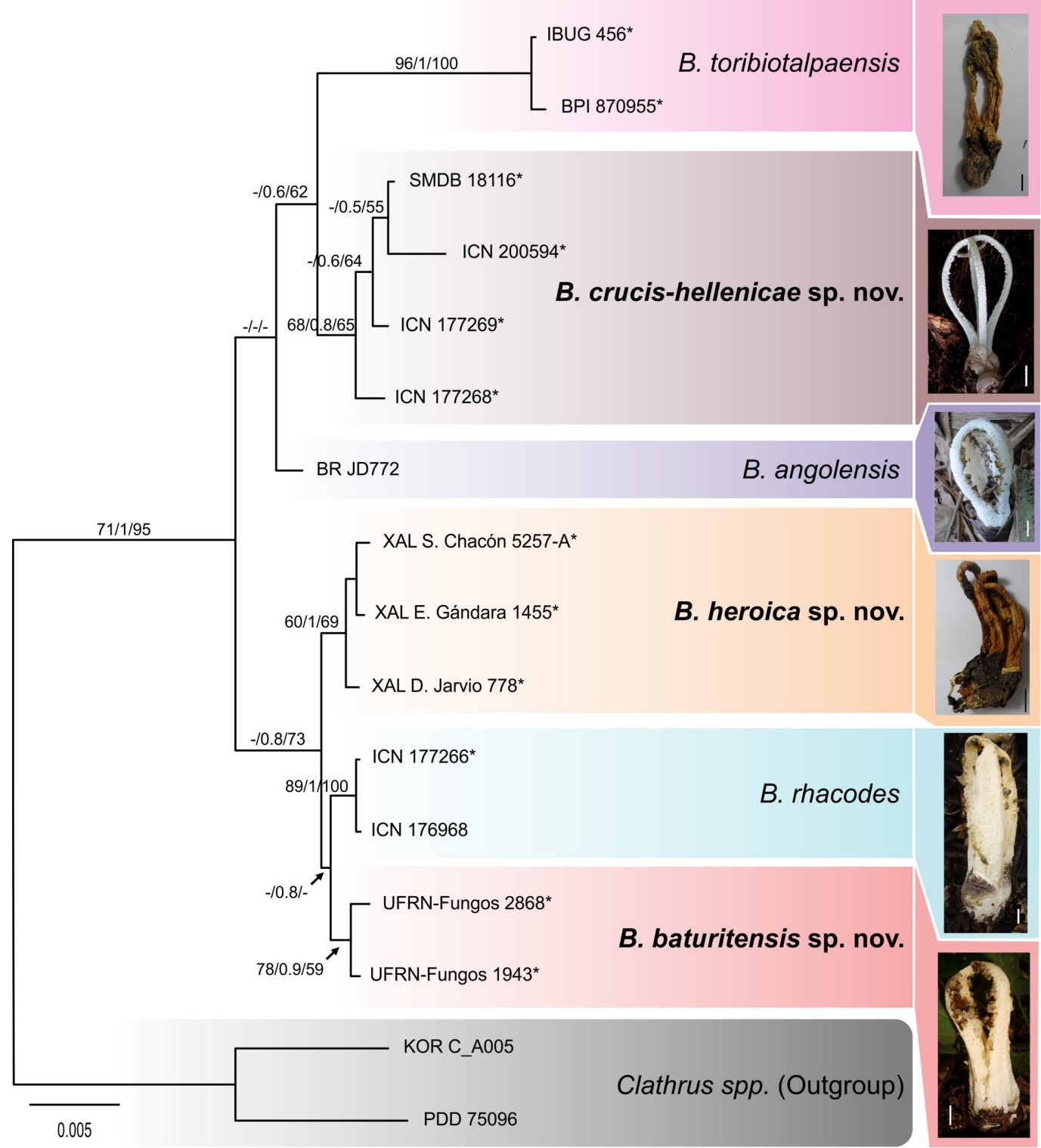

**Fig 1. Bayesian tree from concatenated matrix (ITS1/5.8S/ITS2/LSU/*ATP6*/*RPB2*/*TEF*-1α).** Numbers above branches indicate maximum parsimony bootstrap (MPbs), posterior probability (PP) and maximum likelihood bootstrap (MLbs) values. Asterisks represents type species. To the right, basidiomata of each species. Scale bars = 10 mm.

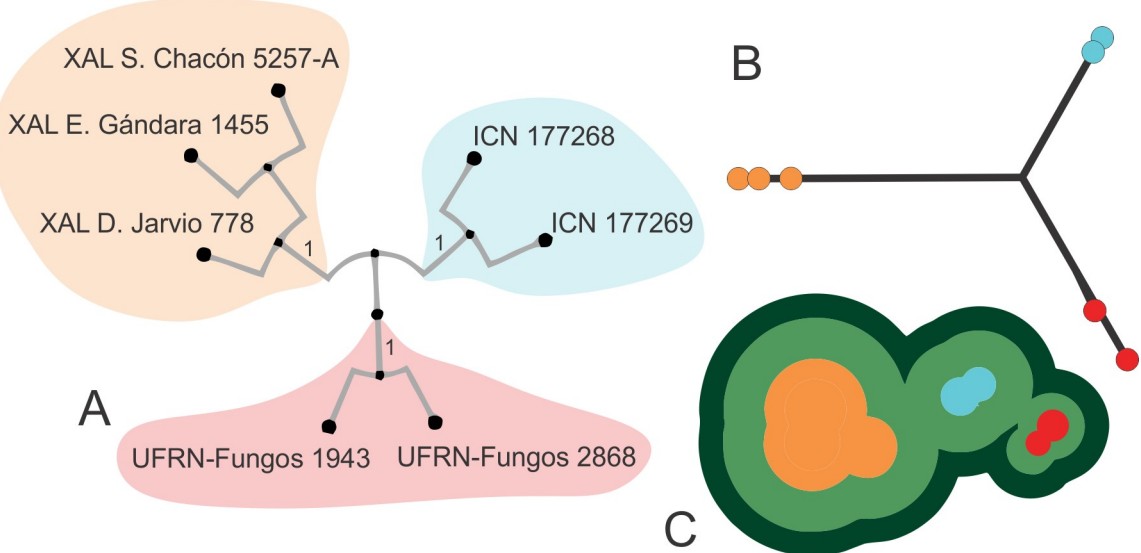

**Fig 2. Species delimitation for *Blumenavia rhacodes*, *B. heroica* sp. nov. and *B. baturitensis* sp. nov.** (A) Phylogenetic tree obtained by Bayesian analysis on concatenated five loci dataset using MrBayes. (B) bPTP PhyloMap [40] based on Bayesian analysis. (C) Topo-phylogenetic representation [41] of Bayesian analysis. *Blumenavia rhacodes* specimens: ICN 176968, ICN 177266; *B. heroica* sp. nov. specimens XAL S. Chacón 5257-A, (paratype), XAL E Gándara 1455 (holotype), XAL D. Jarvio 778 (paratype), *B. baturitensis* sp. nov. specimens: UFRN-Fungos 1943 (paratype), UFRN-Fungos 2868 (holotype).

existing a phylogeographical pattern in *Blumenavia*, which need to be further explored in future studies with better sampling.

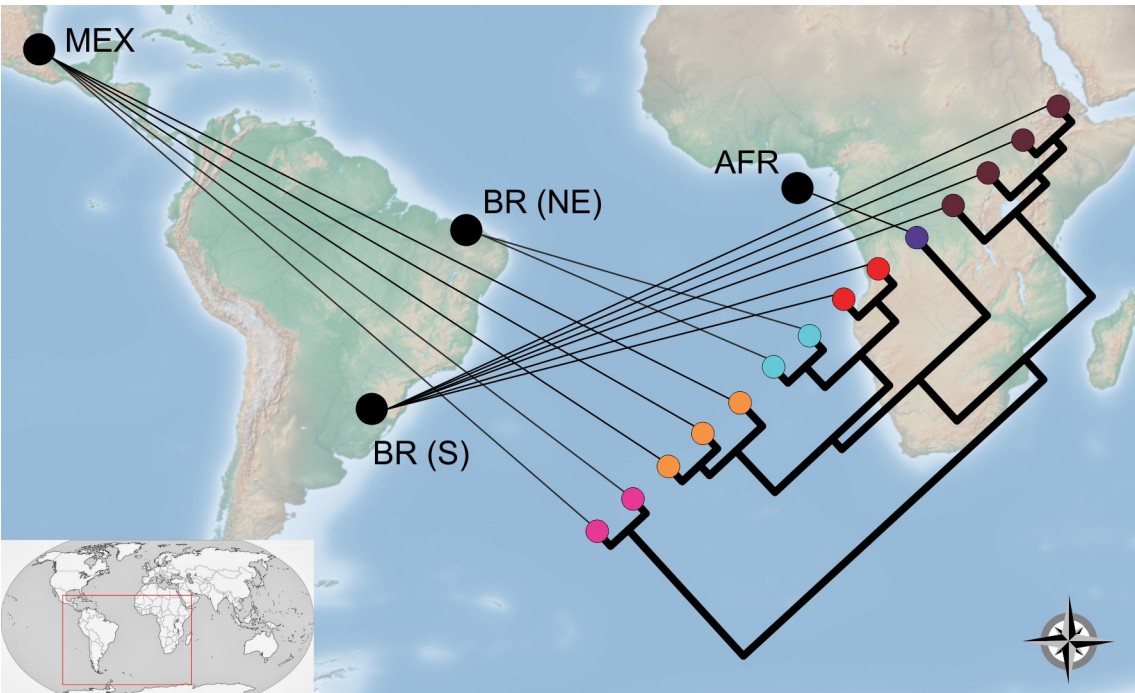

**Fig 3. Geophylogeny proposed for *Blumenavia*.** In nodes: pink *Blumenavia toribiotalpaensis*; orange *B. heroica* sp. nov.; blue *B. baturitensis* sp. nov.; red *B. rhacodes*; purple *B. angolensis*; brown *B. crucis-hellenicae* sp. nov.

## Morphological characters

After all analysis the diagnostic characters for *Blumenavia* genus are macroscopically: length of the basidiomata; color and width of the arms, presence of a groove on the external and internal face of each arm and glebifer position and shape; microscopically: basidiospore size and shape; apical exoperidia hyphal type and wall thickness of rhizomorph hyphae.

Some characters are revealed to be uninformative for species delimitation. Macroscopically are they: number of arms, volva color and shape, rhizomorph and gleba color; microscopically: receptacle, glebifer, basal exoperidia, mesoperidia and endoperidia hyphae length.

In relation to chemical reaction of the microstructures the basidiospores are cyanophilous in all species, and they do not react in 1% Congo red or Melzer's reagent. The hyphae of all parts of the basidiomata are hyaline, and react becoming light red in 1% Congo red, except the hyphae of the apical part of the exoperidia that have a dark brown lumen.

## Taxonomy

**Key to *Blumenavia* species.**   1. White arms ..................................................................................................................... 2

1'. Yellowish white to pale-yellow arms ....................................................................... 4

2. Glebifer as a cross-shaped pyramidal structure well individualized from receptacle apex ....................................................................... *Blumenavia crucis-hellenicae* sp. nov.

2'. Glebifers as a wrinkled tentacular projections structure, or triangular, quadrangular or irregular points projections........................................................................................ 3

3. Basidiospores 3.3–4.0 × 1.4–1.8 μm .................................................... *B. angolensis*

3'. Basidiospores 2.6–3.3 × 1.1–1.5 μm ........................................... *B. usambarensis*

4. Basidimata up to 80mm length and apical exoperidium composed of filamentous and subglobose to ellipsoid hyphae .................................................... *B. heroica* sp. nov.

4'. Basidimata above 80mm length and apical exoperidium composed of filamentous and elongate hyphae or only filamentous ........................................................................... 5

5 Arms are thinned from the middle to the top, torn and twisted glebifers from the middle to the base of the arms ......................................................................................................... 6

5'. Arms even thickness throughout the length, triangular, quadrangular or irregular glebifers regularly spaced along the full length of the arms ....................................................... *B. rhacodes*

6. Outer face of the arms with a marked longitudinal groove ........................ *B. toribiotalpaensis*

6'. Outer face of the arms without a marked longitudinal groove ............ *B. baturitensis* sp. nov.

**Blumenavia angolensis (Welw. & Curr.) Dring, Kew Bull. 35(1): 53 (1980), Fig 4.**

- *Laternea angolensis* Welwitsch & Currey, Trans. Linn. Soc. London 26(1): 286 (1870)

- *Clathrus angolensis* (Welw. & Curr.) Fischer, Jahrb. Bot. Gart. Mus. Berlin 4: 70 (1886)

- *Colonnaria angolensis* (Welw. & Curr.) Fischer in Engler & Prantl, Nat. Pflanzenfam., 2 Aufl. 7a: 85 (1933)

*Holotype*. ANGOLA. Pungo Andongo, pr. Catete, 10 Dec. 1856, Welwitsch 120 (K! Welwistch and Currey [5]: Table 17, Fig 7).

*Description*. Since specimens were not located, this description is based on Degreef et al. [8], and on the color photography included in Desjardin and Perry [11], see Fig 4A and 4B. Expanded basidiomata 80–85 mm length × 35–40 mm wide. *Exoperidium* white

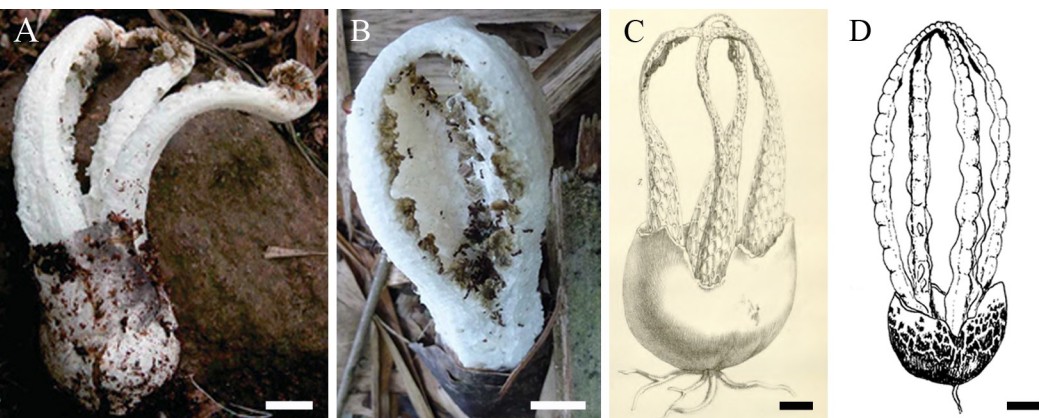

**Fig 4. *Blumenavia angolensis*.** (A) Fresh basidiomata from São Tomé, reproduced from Degreef et al. [8] Fig 2C, ©
*Publications Scientifiques du Muséum national d'Histoire naturelle*, Paris. (B) Fresh basidiomata from São Tomé, reproduced
from Desjardin and Perry [11] Fig 6, © Desjardin and Perry. (C) Basionym *Laternea angolensis*, reproduced from Welwitsch
and Currey [5] Tab. 17, Fig 7. (D) Illustration reproduced from Dring [4] Fig 15F, © *Board of Trustees of the Royal Botanic
Gardens*, *Kew*. (A–D) bar = 10 mm.

(N00Y00M00) smooth base, cracked into brownish gray ($N_{60}Y_{00}M_{00}$–$N_{90}Y_{00}M_{00}$) scales
above. Receptacle 4 arms united above, free in the base, white ($N_{00}M_{00}C_{00}$), arms even thick-
ness throughout the length, without grooves. Membranous glebifers, adhered to the anterolat-
eral angles of the arms, triangular, quadrangular or irregular, regularly spaced along the full
length of the arms, glebifers covered with glebal mass. Basidiospores ellipsoid, (3.2–)3.3–3.6–4
(–4) × (1.3–)1.4–1.6–1.8(–1.9) μm, Q = (1.88)1.89–2.25–2.61(–2.68).

*Habit and habitat*. Epigeous and solitary, saprotrophic in secondary forest.

*Known distribution*. Afrotropic: Angola, Pungo Andongo [5]; São Tomé and Príncipe, São
Tomé [8,11].

*Comments*. Specimens of this species were not analyzed because recent collections were not
located. We contacted Dr. Jerome Degreef (Belgium) to study the specimen (Fig 4A) described
in Degreef et al. [8], since there is an available LSU sequence in GenBank; Dr. Degreef con-
firmed to us that the specimen was not located, nor the DNA isolate, thus it was not possible to
obtain sequences from more loci. The specimen from Desjardin and Perry [11] was not con-
served, because the authors did not collect the specimens, they took only pictures of them (Fig
4B).

Dring [4] based on the image protologue (Fig 4C) and specimens from Brazil and Trinidad
and Tobago made an illustration for *B. angolensis* (Fig 4D). Degreef et al. [8] based their analy-
sis in one specimen from Sao Tome and Principe (Fig 4A). Dring [4] and Degreef et al. [8] sep-
arated *B. angolensis* from *B. rhacodes* according to the distribution of the glebifer, and the
color of their basidiomata. However, our study confirms the importance of color for distin-
guishing the basidiomata of these species, but not the glebifer distribution, since both species
have their glebifers distributed along the arms.

Desjardin and Perry [11] mentioned Degreef et al. [8] for the description of the photo-
graphed basidiomata from Sao Tome and Principe. This photograph (Fig 4B) shows the glebi-
fers regularly spaced along the length of the arms. Although this character differs from the
protologue [5] (Fig 4C), and from the description provided by Degreef et al. [8], we adopted
this image (Fig 4B) to describe this character and exemplify the species *B. angolensis* in the
phylogenetic tree (Fig 1). Degreef et al. [8] probably analyzed an old basidioma with the arms
not joined at the apex. Also, because of zoochoric dispersion, insects may have consumed

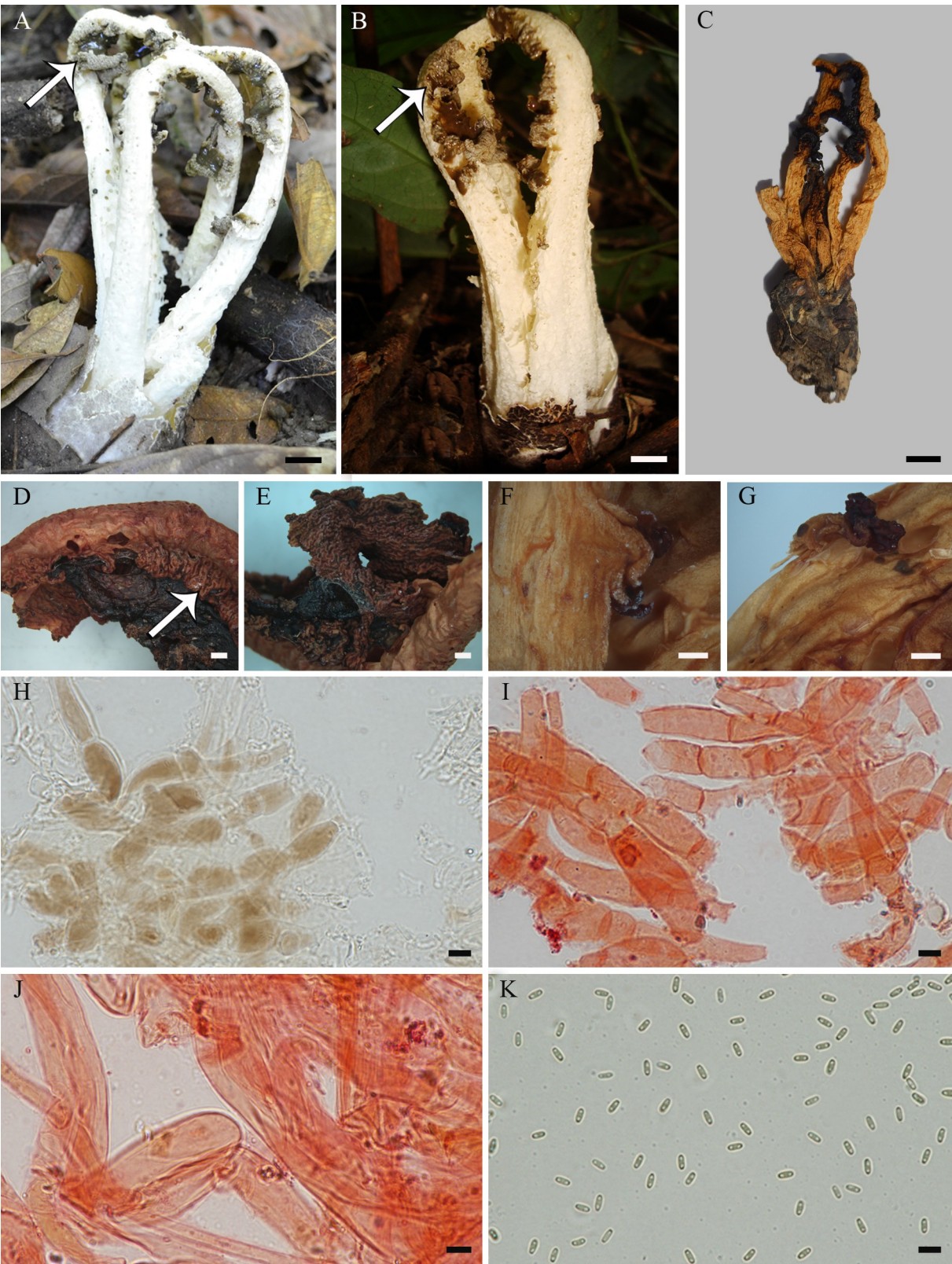

**Fig 5.** ***Blumenavia baturitensis* sp. nov.** (A) Fresh basidiomata, UFRN-Fungos 2868, holotype. (B–I) UFRN-Fungos 1943, paratype: (B) Fresh basidiomata; (C) Dry basidiomata; (D) Wrinkled tentacular projections of glebifer; (E) Glebifer projections forming a larger projection leaving

a fusiform space between them; (F, G) Torn and twisted glebifers in arms; (H) Apical exoperidium hyphae in 5% KOH; (I) Apical exoperidium hyphae in 1% Congo red. (J, K) UFRN-Fungos 2868, holotype: (J) Rhizomorph hyphae in 1% Congo red; (K) Basidiospores in Melzer's reagent. (A, B, D) Arrows indicating membranous wrinkled, tentacled glebifers. (A–C) bar = 10 mm; (D, E) bar = 1 mm; (F, G) bar = 0.5 mm; (H–K) bar = 5 μm.

parts of the basidiomata, leaving only triangular projections without gleba along the arms, as can be seen in Fig 4A. *Blumenavia angolensis* has white arms and ellipsoid basidiospores, (3.2–)3.3–4.0 × (1.3–)1.4–1.8(–1.9) μm. *Blumenavia usambarensis* and *B. crucis-hellenicae* sp. nov. also has white arms; however, *B. usambarensis* has smaller basidiospores (2.6–3.3 × 1.1–1.5 μm), and *B. crucis-hellenicae* sp. nov. has a cross-shaped glebifer.

***Blumenavia baturitensis*** **Melanda, M.P. Martín & Baseia, sp. nov., [Fig 5](), MycoBank MB 831134.** *Diagnosis. Blumenavia baturitensis* sp. nov. has a whitish yellow receptacle, like *B. rhacodes*, *B. heroica* sp. nov., and *B. toribiotalpaensis*, but differs by the lack of a groove in the outer face of the arms and the glebifer formed by wrinkled, tentacle (finger-like) projections. *Blumenavia baturitensis* sp. nov. differs also from *B. rhacodes* by the arms thinned from the middle to the top of the basidiomata.

*Holotype.* BRAZIL. Ceará, Guaramiranga, APA do Maciço de Baturité, Trilha dos Veadeiros, 04°15′15.2″S 38°55′49.28″W, 844 m, 30 Jun. 2016, leg. G.C.S. Melanda 15, A. A., Lima (UFRN-Fungos 2868!, ITS nrDNA, LSU nrDNA, *ATP*6, *RPB*2 and *TEF*-1α GenBank sequences: MG817725, MG817733, MH061924, MH061933, MH061942).

*Etymology.* The name of the species refers to the type locality, located in Baturité microregion, Ceará, Brazil.

*Description.* Expanded basidiomata 91–119 mm length × 35–55 mm wide. Peridium composed of three layers. Exoperidium light yellow to dark yellow ($N_{10}C_{00}Y_{30}$–$N_{10}C_{00}Y_{60}$) smooth base, cracked into brownish gray ($N_{60}Y_{00}M_{00}$–$N_{90}Y_{00}M_{00}$), dark brown ($N_{90}Y_{80}M_{40}$) or black ($N_{99}Y_{30}M_{00}$) scales above. Single, branched, white rhizomorph ($N_{00}M_{00}C_{00}$). Receptacle 4 to 5 arms united above, free in the base, whitish yellow ($Y_{70}M_{10}C_{00}$) or white ($N_{00}Y_{00}C_{00}$) near the volva becoming yellowish ($N_{00}Y_{50}M_{00}$) at the apex, the arms are thinned from the middle to the top of the basidiomata, without grooves ([Fig 5A–5C]()). Membranous glebifers adhered to the anterolateral angles of the arms, formed by wrinkled, tentacled (finger-like) projections, grouped parallel and regularly spaced in the upper half of the basidiomata ([Fig 5A–5D](), see the arrows), in some points near the apex these projections coming from each side of the arm are joined and form a larger projection leaving a fusiform space between them ([Fig 5E]()), this basidiomata have torn and twisted projections from the middle to the base of the arms ([Fig 5F and 5G]()), glebifers covered with glebal mass.

Apical exoperidium composed of filamentous, branched hyphae, regularly septate, 2.8–8.7 μm diam., hyaline, some of them with dark brown lumen ([Fig 5H]()), straight and regular walls, 0.3–0.8 μm thick ([Fig 5I]()). Rhizomorphs composed of filamentous hyphae, branched, irregularly septate, 1.4–5.4 μm diam., hyaline, straight and regular walls, some of them slightly winding, 0.3–0.9 μm thick ([Fig 5J]()). Basidiospores cylindrical, 2.9–4.2 × 1.1–2.0 μm ($x = 3.5 ± 0.2 × 1.5 ± 0.1$, Qm = 2.42 ± 0.18), smooth, with an inner gutule at each end of the length, hyaline in KOH and Congo red, inamyloid ([Fig 5K]()) and cyanophilous.

*Habit and habitat.* Epigeous and solitary, lignicolous; in 'Brejo de altitude' forest (Northeastern fragment of Atlantic Forest), on decaying wood and on litter.

*Known distribution.* Neotropic: Brazil, Ceará.

*Additional material examined.* BRAZIL. Ceará, Guaramiranga, APA do Maciço de Baturité, Trilha dos Veadeiros, 04°15′18.8″S; 38°55′52.3″W, 865 m, 7 Jul. 2012, leg. A.C.M. Rodrigues, T.L.D. Silva (**paratype**: UFRN-Fungos 1943, ITS nrDNA, LSU nrDNA, *ATP*6, *RPB*2 and *TEF*-

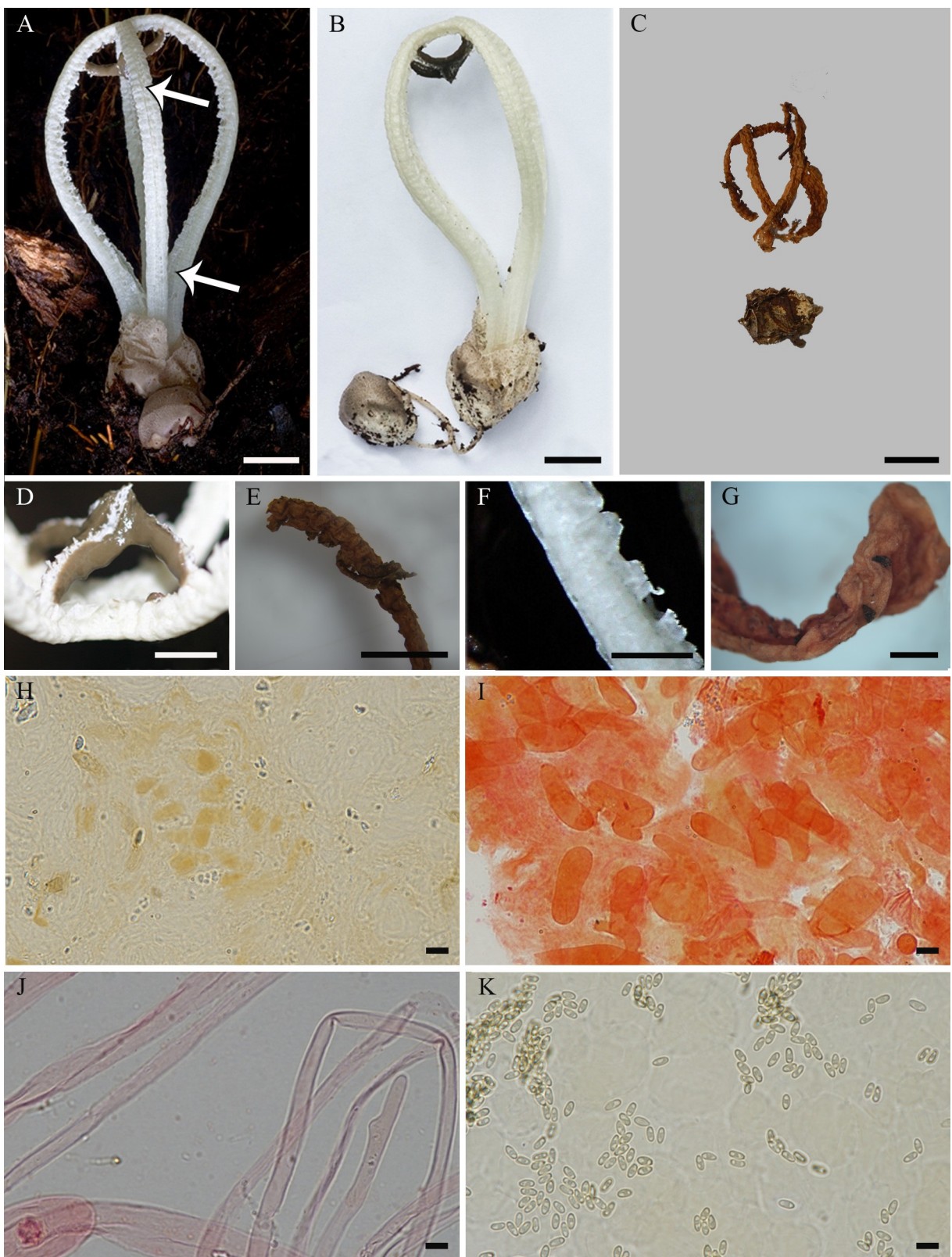

**Fig 6. *Blumenavia crucis-hellenicae* sp. nov.** (A, B) Fresh basidiomata, SMDB 18116, holotype. (C) Dry basidiomata, ICN 177268, paratype. (D, E) Glebifer as a cross-shaped pyramidal structure: (D) Fresh, SMDB 18116, holotype; (E) Dry stuck in one arm, ICN 177268, paratype. (F,

G) Glebifers like teeth along the extension of the arms: (F) Fresh, SMDB 18116, holotype; (G) Dry, ICN 177268, paratype. (H–K) ICN 177268, paratype: (H) Basidiospores in Melzer's reagent; (I) Rhizomorph hyphae in 1% Congo red; (J) Apical exoperidium hyphae in 1% Congo red; (K) Apical exoperidium hyphae in 5% KOH. (A) Arrows indicating the longitudinal outer and inner grooves. (A–E) bar = 10 mm; (F, G) bar = 5 mm; (H–K) bar = 5 μm.

1α GenBank sequences: MG817726, MG817734, MH061925, MH061934, MH061943, respectively).

*Comments.* Specimens of this new species (collection UFRN-Fungos 1943) were identified based on morphological characters alone in Rodrigues and Baseia [9] and Trierveiler-Pereira et al. [10] as *B. angolensis* and *B. rhacodes*, respectively. However, *B. baturitensis* sp. nov. differs from *B. angolensis* by having larger basidiomata (91–119 mm), yellowish and thick arms, tapering from the middle of the basidiomata to the apex (Fig 5A and 5B), and glebifers as wrinkled tentacular projections (Fig 5D). While *B. rhacodes* has groove in the outer face of the arms, and the glebifer shape different, as also confirmed by molecular data. The new species shows similar characteristics to *B. toribiotalpaensis*; however, the glebifer projections are tentacular (Fig 5D) in *B. baturitensis* sp. nov. and lacerated in *B. toribiotalpaensis* (Fig 10F) in the upper part of basidiomata. In our phylogenetic tree (Fig 1), *B. baturitensis* sp. nov. forms a sister clade with *B. rhacodes*, while *B. toribiotalpaensis* is closer to *B. crucis-hellenicae* sp. nov. and *B. angolensis*.

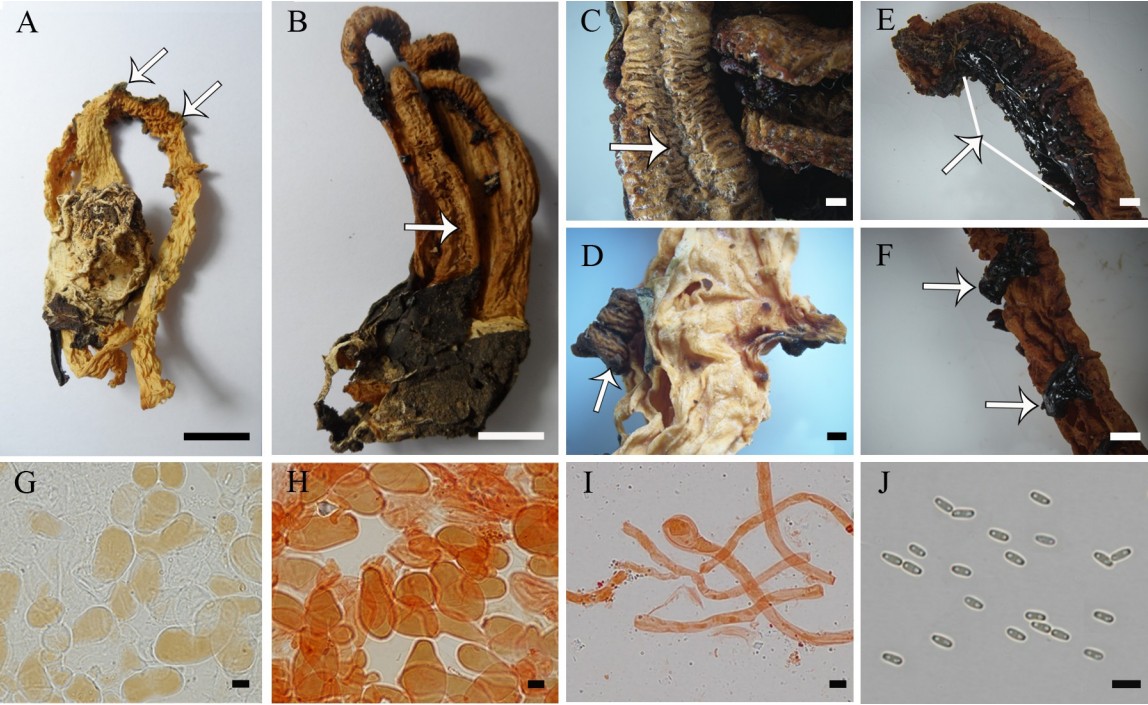

**Fig 7. *Blumenavia heroica* sp. nov.** (A–B) Dry basidiomata: (A) S. Chacón 5257-A, paratype; (B) XAL R. Medel 672, paratype. (C) Groove in the outer face of the arm, XAL D. Jarvio 370, paratype. (D) Glebifers forming by parallel lines curving inward, S. Chacón 5257-A, paratype. (E, F) XAL R. Medel 672, paratype: (E) Glebifers forming by parallel lines curving inward not sparse in the upper half of the basidiomata; (F) Spaced glebifers. (G, H) XAL E. Gándara 1455, holotype: (G) Apical exoperidium hyphae in 5% KOH; (H) Apical exoperidium hyphae in 1% Congo red. (I) Rhizomorph hyphae in 1% Congo red, XAL S. Chacón 4475, paratype. (J) Basidiospores in 5% KOH, XAL E. Gándara 1455, holotype. (A, D, E, F) Arrows indicating glebifers. (B, C) Arrows indicating longitudinal outer groove. (A, B) bar = 10 mm; (C, E, F) bar = 1 mm; (D) bar = 5 mm; (G–J) bar = 5 μm.

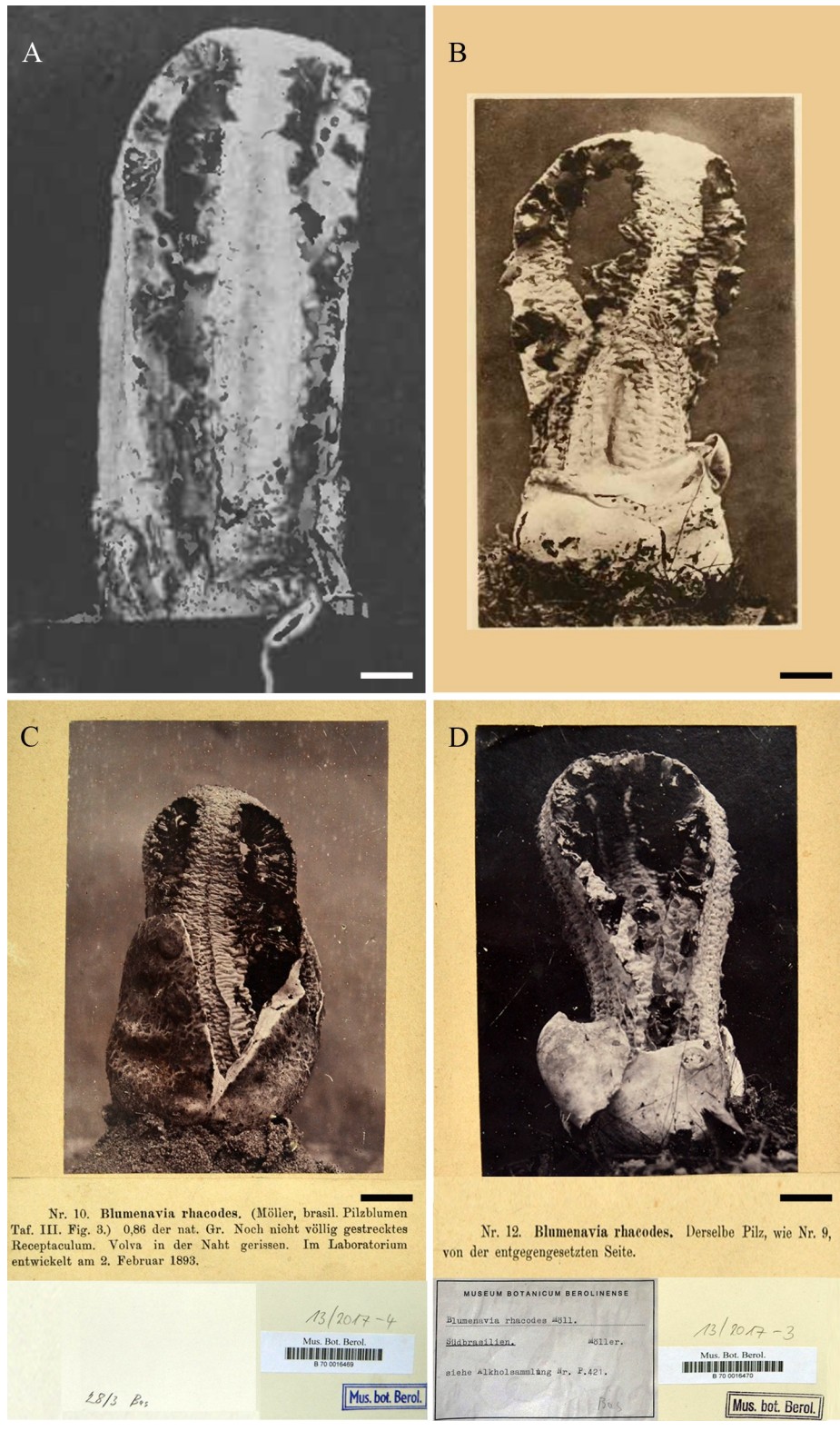

**Fig 8. *Blumenavia rhacodes* photographs.** (A–C) Reproduced from Möller [3], Tab. III: (A) Fig 1A sintype; (B) Fig 2 sintype; (C) Fig 3, lent from B Herbarium B700016469, sintype. (D) Lent from B Herbarium, B700016470. (A–D) bar = 10 mm.

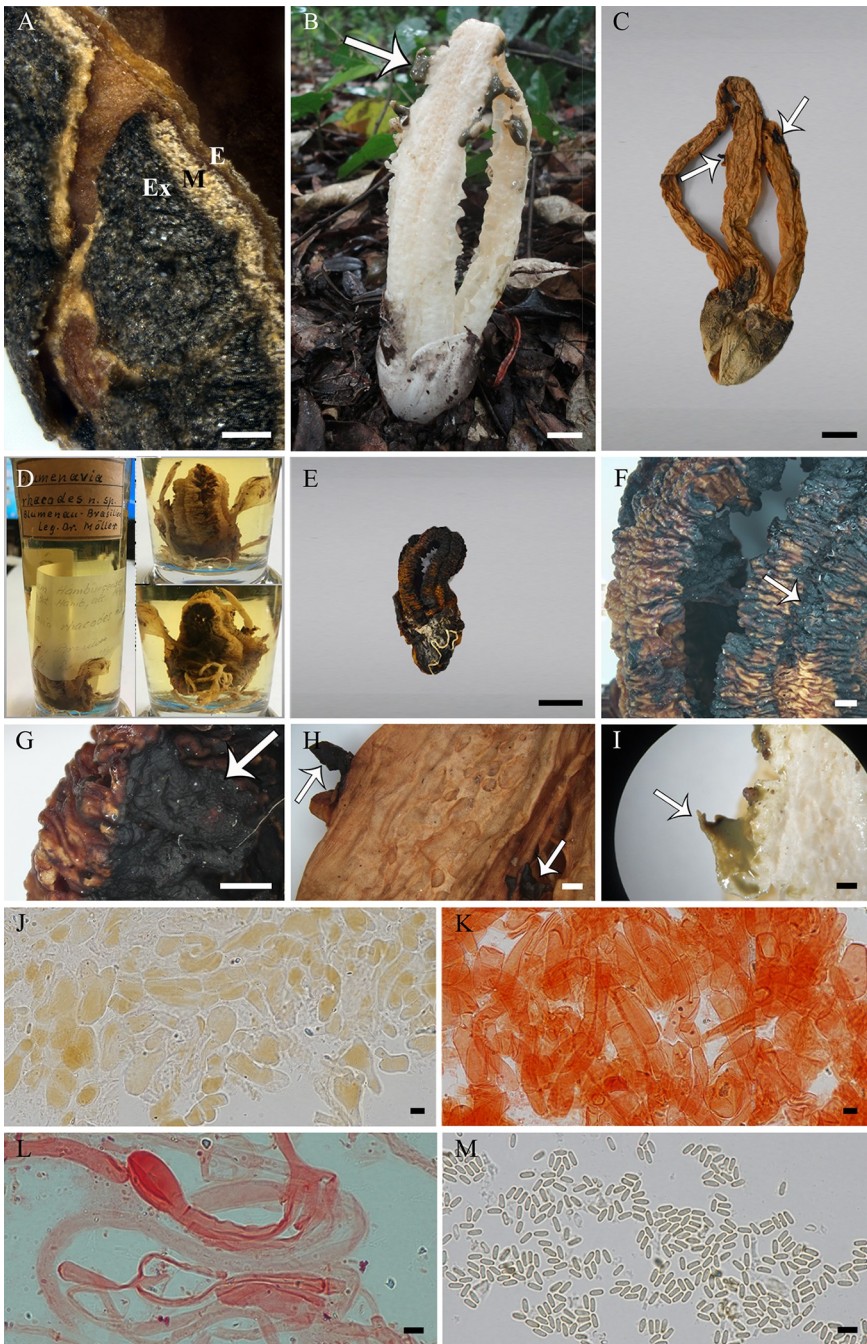

**Fig 9. *Blumenavia rhacodes*.** (A) Peridium layers, ICN177267: Ex = exoperidium, M = mesoperidium,
E = endoperidium. (B–C) ICN 177266, epitype: (B) Fresh basidiomata; (C) Dry basidiomata. (D) Basidiomata
preserved in alcohol, HBG 024640. (E–G) B700016478, lectotype: (E) Dry basidiomata; (F) Groove in the arm. (G) Dry
glebifer. (H–K) ICN 177266, epitype. (H) Dry glebifer; (I) Fresh glebifer; (J–K) ICN 177266, epitype: (J) Apical
exoperidium hyphae in 5% KOH; (K) Apical exoperidium hyphae in 1% Congo red. (L) Rhizomorph hyphae in 1%
Congo red, PACA 12552. (M) Basidiospores in 5% KOH, PACA 12550. (B, C, G, H, I) Arrows indicating glebifers. (F)
Arrow indicating the longitudinal outer groove. (A) bar = 0.2 mm; (B, C, E) bar = 10 mm; (F–I) bar = 1mm; (J–M)
bar = 5 μm. Photographs: (B, I) sent by Fazolino E. P.; (D) sent by Matthias Schultz.

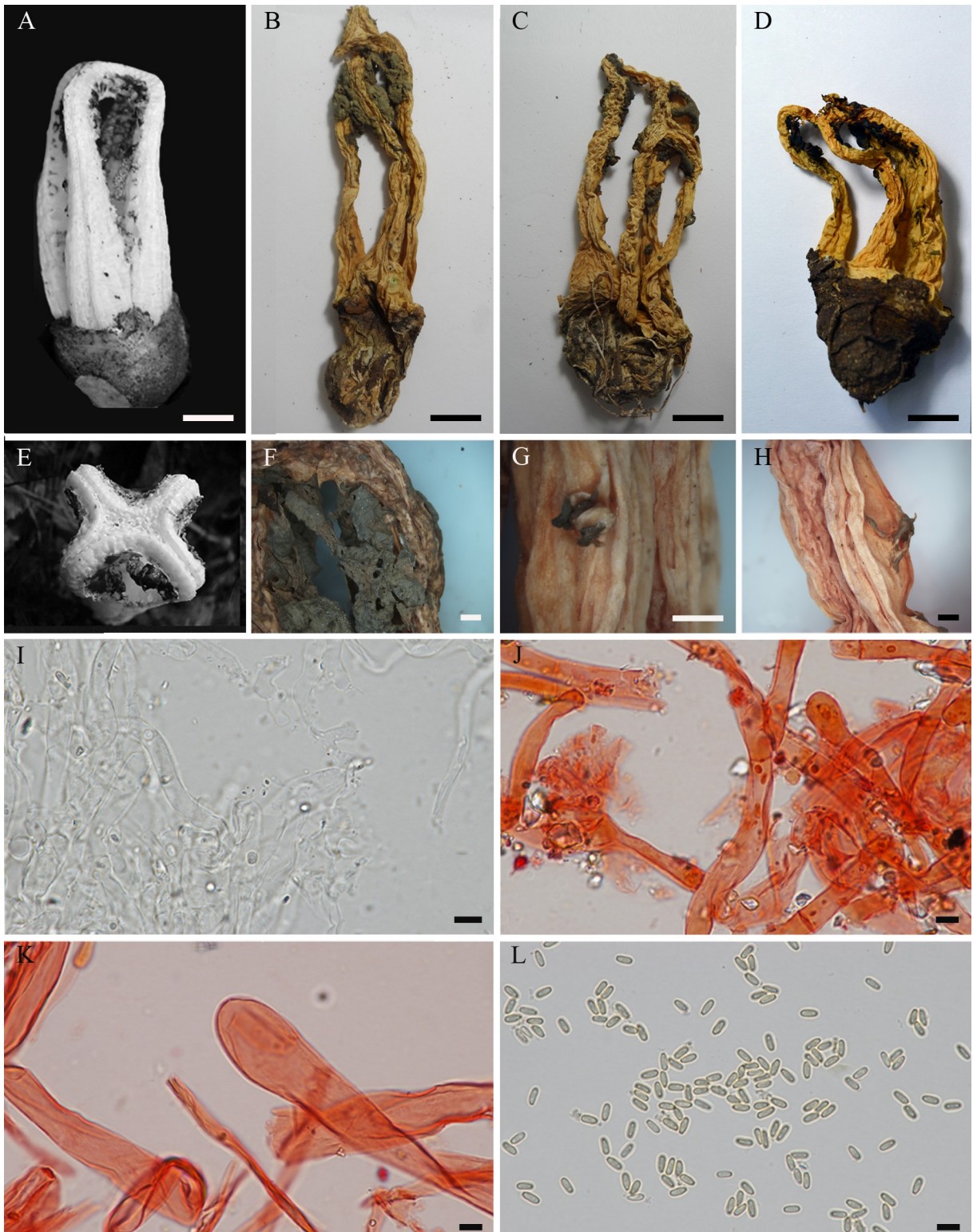

**Fig 10. *Blumenavia toribiotalpaensis.*** (A) Fresh basidiomata, black and white image reproduced Vargas-Rodriguez and Vázquez-García [7] Fig 3, © Vargas-Rodriguez and Vázquez-García. (B, C) Dry basidiomata, BPI 870955, holotype. (D) Dry basidiomata, IBUG 456, paratype. (E) Top view of fresh basidiomata show groove in the arm, black and white image reproduced from Vázquez-García [7] Fig 5, © Vargas-Rodriguez and Vázquez-García. (F) Lacerated projections of glebifer, BPI 870955, holotype. (G, H) Torn and twisted points of glebifer, BPI 870955, holotype. (I, J) IBUG 454, paratype: (I) Apical exoperidium hyphae in 5% KOH; (J) Apical exoperidium hyphae in 1% Congo red. (K)

Rhizomorph hyphae in 1% Congo red, IBUG 240a, isotype. (L) Basidiospores in 5% KOH, IBUG 456, paratype. (A–B) bar = 10 mm; (F–H) bar = 1 mm; (I–L) = 5 μm.

***Blumenavia crucis-hellenicae*** **G. Coelho, Sulzbacher, Grebenc & Cortez, sp. nov., Fig 6,** **MycoBank MB 831344.** *Diagnosis. Blumenavia crucis-hellenicae* sp. nov. differs from all other species in the genus by its glebifer as a cross-shaped pyramidal structure from the receptacle apex, with grooves in the outer and inner face of the white arms, and elongated basidiospores with a Qm = 1.93 ± 0.16.

*Holotype*. BRAZIL. Rio Grande do Sul, São Francisco de Paula, Potreiro Velho, Centro de Pesquisa e Conservação da Natureza Pró-Mata (CPCN Pró-Mata), 27 Jun. 2010, leg. G. Coelho & M.A. Sulzbacher (SMDB 18116, ITS nrDNA, LSU nrDNA, *ATP*6, *RPB*2 and *TEF*-1α Gen-Bank sequences: LN875253, MK958819, MK975457, MK975458, LN875257, respectively).

*Etymology*. From the Latin *crucis* (cross) + *hellenicae* (from Greece, *graeca*); referring to the form of glebifer like an iconographic Greek cross (two perpendicular bars of the same length, *crux quadrata*).

*Description*. Expanded basidiomata 40–90 mm de length × 27–40 mm wide. Peridium composed of three layers. Exoperidium white to dark yellow ($N_{00}Y_{00}M_{00}$–$N_{10}C_{00}Y_{70}$) and smooth base, cracked into light to dark brown ($N_{30}Y_{60}M_{30}$–$N_{90}Y_{80}M_{40}$) scales above. Single, branched, white rhizomorph ($N_{00}M_{00}C_{00}$). Receptacle 3 to 4 arms united above, free in the base, white ($N_{00}M_{00}C_{00}$), arms even thickness throughout their length, outer and inner face with a marked longitudinal groove (Fig 6A–6C, arrows in A indicating the grooves). Membranous glebifers as a cross-shaped pyramidal structure well individualized from receptacle's apex, but turned on its base, formed by four sickle-like pieces starting from the inner side of each arm (slightly distant from arm junction) to joining at the center of receptacle, they are thinner at the arm insertion and enlarged towards the center (Fig 6D and 6E), also with glebifers like teeth along the extension of the arms (Fig 6F and 6G), glebifers covered with glebal mass.

Apical exoperidium composed of filamentous and elongated hyphae (Fig 6H and 6I). Filamentous hyphae branched, regularly septate, 2.7–6.7 μm diam., hyaline, some of them with dark brown lumen (Fig 6H), straight and regular walls, 0.2–0.6 μm thick. Elongated hyphae look like the filamentous hyphae separated in the septa, 6.8–28.5 × 3.5–14.2 μm, yellowish brown lumen, straight and regular walls, 0.3–1.3 μm thick. Rhizomorphs composed of filamentous hyphae, branched, irregularly septate, 1.9–7.1 μm diam., hyaline, straight and regular walls, 0.2–1.0 μm thick (Fig 6J). Basidiospores elongated (oblong), 3.2–4.1 × 1.5–2.2 μm ($x$ = 3.7 ± 0.2 × 1.9 ± 0.1, Qm = 1.93 ± 0.16), smooth, with an inner gutula at each end of the length, hyaline in KOH and Congo red, inamyloid (Fig 6K) and cyanophilous.

*Habit and habitat*. Epigeous and solitary, saprotrophic in *Araucaria* forest, and mixed ombrophilous native forest, on decaying wood and on litter.

*Known distribution*. Neotropic: Brazil, Paraná and Rio Grande do Sul.

*Additional material examined*. BRAZIL. Paraná, Antonina, Reserva Natural do Rio Cachoeira, 17 Aug. 2005, leg. A.A.R. de Meijer 4339 (**paratype:** MBM Meijer 4339, as *Blumenavia* sp.). Rio Grande do Sul, São Francisco de Paula, Floresta Nacional de São Francisco de Paula (FLONA), 29°26′53″S 50°35′1″W, 08 Feb. 2014, leg. C.R. Alves 143 (**paratype:** ICN 177268, as *B. angolensis*, LSU nrDNA, *ATP*6, *RPB*2 and *TEF*-1α GenBank sequences: MG817727, MH061920, MH061926, MH061935, respectively); *ibid*. 13 Apr. 2014, leg. A. C. Magnano 1049 (**paratype:** ICN 177269, as *B. angolensis*, ITS nrDNA, LSU nrDNA, *ATP*6 and *RPB*2 GenBank sequences: MG817717, MG8177280, MH061921, MH061927, respectively). Rio Grande do Sul, São Francisco de Paula, CPCN Pró-Mata, 29°28′43.8″S 50°10′23″W, 500

m, 26 May 2015, leg. Fazolino E. P. 533 (**paratype**: ICN 200594, ITS nrDNA and TEF-1α Gen-Bank sequences: LN875254, LN875258, respectively).

*Comments*. Trierveiler-Pereira et al. [10] identified the collections ICN 177268 and ICN 177269 (studied here) as *B. angolensis*, but *B. crucis-hellenicae* sp. nov. differs from *B. angolensis*, and from the other species in the genus, by the elongated (oblong) basidiospores (Fig 6K), with a Qm = 1.93 ± 0.16 unlike the other species that have a range between Qm = 2.22 ± 0.15 to Qm = 2.42 ± 0.20. Moreover, *B. crucis-hellenicae* sp. nov. has a cross-shaped glebifer (Fig 6D), and grooves in the inner part of the arms (Fig 6A and 6B). Molecular analyses do not confirm the existence of a cluster with white colored species as mentioned by other authors [4,10] since the white species *Blumenavia crucis-hellenicae* sp. nov. and *B. angolensis* shares a common ancestor with *B. toribiotalpaensis*, one of the yellowish species (Fig 1). Branch supports and morphological analyses indicate *B. angolensis*, *B. crucis-hellenicae* sp. nov. and *B. toribiotalpaensis* as three independent species.

**Blumenavia heroica** Melanda, Baseia & M. P. Martín, **sp. nov., Fig 7, MycoBank MB 831136.** *Diagnosis. Blumenavia heroica* sp. nov. is the only species in this genus with filamentous and subglobose to ellipsoid apical exoperidium hyphae, unlike filamentous and elongate hyphae in *B. rhacodes* and *B. angolensis* and only filamentous in *B. toribiotalpaensis* and *B. baturitensis* sp. nov. *Blumenavia heroica* have up to 80 mm length, in opposition the bigger yellowish species of *Blumenavia*.

*Holotype*. MEXICO. Veracruz, Xalapa, Parque Ecológico "Francisco Javier Clavijero", 19 Aug. 2005, leg. E. Gándara (XAL E. Gándara 1455! as *Collonaria*, ITS nrDNA, *RPB2* and *TEF-*1α GenBank sequences: MG817721, MH061930, MH061939, respectively).

*Etymology*. Named in honor to the type locality, Veracruz, also known as Heroica Veracruz.

*Description*. Expanded basidiomata 45–80 mm length × 13–30 mm wide. Peridium composed of three layers. Exoperidium all dark grey, almost black ($N_{99}Y_{10}M_{00}$) smooth, or white ($N_{00}Y_{00}M_{00}$) smooth base, cracked into light to dark brown ($N_{30}Y_{60}M_{30}$–$N_{90}Y_{80}M_{40}$) scales above. Receptacle 3 to 4 arms united above, free in the base, whitish yellow to pale yellow ($Y_{70}M_{10}C_{00}$–$Y_{70}M_{20}C_{10}$), the arms even thickness throughout its length (Fig 7A) or are thinner from the middle to the top of the basidiomata (Fig 7B) with a marked longitudinal groove in the outer face (Fig 7B and 7C, see the arrows). Membranous glebifers adhered to the anterolateral angles of the arms. Glebifers composed by parallel lines curving inward (Fig 7A and 7D, see the arrows) forming triangular, quadrangular or irregular points projections regularly spaced along the length of the arms or parallel lines curving inward not sparse in the upper half of the basidiomata (Fig 7E, see the arrow) with triangular to quadrangular spaced projections to the base (Fig 7F, see the arrows), glebifers covered with glebal mass.

Apical exoperidium composed of filamentous and subglobose to ellipsoid hyphae (Fig 7G and 7H). Filamentous hyphae, branched, regularly septate, 3.0–10.0 μm diam., hyaline, straight and regular walls, 0.3–1.0 μm thick. Subglobose to ellipsoid hyphae, 5.6–30.6 × 3.8–21.5 μm, with dark brown lumen, straight and regular walls, 0.3–1.2 μm thick. Rhizomorphs composed of filamentous hyphae, branched hyphae, irregularly septate, 1.4–8.0 μm diam., hyaline, straight and regular walls, 0.2–1.1 μm thick (Fig 7I). Basidiospores cylindrical, 2.8–4.4 × 1.1–2.1, μm, ($x$ = 3.7 ± 0.2 × 1.6 ± 0.1, Qm = 2.35 ± 0.20), smooth, with an inner gutule at each end of the length, hyaline in KOH (Fig 7J) and Congo red, inamyloid and cyanophilous.

*Habit and habitat*. Epigeous and solitary, saprotrophic in mesophilic mountain forest.

*Known distribution*. Neotropical: Mexico, Veracruz.

*Additional material examined*. MEXICO. Veracruz, Xalapa, Parque Ecológico "Francisco Javier Clavijero", 13 Jun. 1986, leg. D. Jarvio (**paratype**: XAL D. Jarvio 778 as *Collonaria*, ITS nrDNA GenBank sequence: MG817722); *ibid*. 09 June 2000, leg. S. Chacón (**paratype**: XAL S. Chacón 5257-A as *Colonnaria*, ITS nrDNA, LSU nrDNA, *RPB2* and *TEF-*1α GenBank

sequences: MG817720, MG817731, MH061929, MH061938, respectively); *ibid*. 27 Sept. 1998, leg. R. Medel (**paratype**: XAL R. Medel 672, as *Collonaria collumnata*); *ibid*. 09 Nov. 1999, leg. D. Jarvio (**paratype**: XAL D. Jarvio 370 as *Collonaria*). MÉXICO. Veracruz, Xalapa, in a gardens house to the North of the city, 1200 m, 14 Jan. 1991, leg. S. Chacón (**paratype**: XAL S. Chacón 4475 as *Blumenavia angolensis*).

*Comments*. Calonge et al. [12] cited a number of collections under *Blumenavia rhacodes*: IBUG 240a, here designate as *B. toribiotalpaensis*, as well as those located in XAL herbarium (X. Madrigal 4535 E. Gándara 1455, R. Medel 654, J. C. Anel 410, F Tapia 712, S. Chacón 5107, R. Medel 672, S. Chacón 5257-A, D. Jarvio 370, D. Jarvio 778, R. Farias, R. Medel 662, S. Chacón 4475), in which some of them here are considered as the new species *B. heroica* sp. nov. *Blumenavia heroica* sp. nov. has a large macromorphological variety, with basidiomata with varying sizes (45–80 mm), besides presenting arms and glebifers with distinct forms. However, in the micromorphological analyses, these specimens analyzed here share apical exoperidial hyphae with of the same pattern (filamentous or globose) and the same cylindrical basidiospore dimensions. Confirming microscopic studies, phylogenetic analyses demonstrated that these specimens belong to a single well-supported clade (PP = 0.9). That clade shares a common ancestor with *B. rhacodes* and *B. baturitensis* sp. nov., all these three species have a yellowish basidioma, but *B. heroica* sp. nov. differs in the apical exoperidia hyphae and in the basidiomata height. *Blumenavia toribiotalpaensis* also differs from *B. heroica* sp. nov. in that characters mentioned for *B. rhacodes* and *B. baturitensis* sp. nov., and in the glebifer shape.

**Blumenavia rhacodes Möller, Bot. Mitt. Trop.: 57 (1895), Figs 8 and 9.**

- *Laternea rhacodes* (Möller) Lloyd, Synopsis of the known phalloids (7): 50 (1909)

*Lectotype (designated here)*. BRAZIL. Santa Catarina, Blumenau, leg. Möller (B700016478!).

*Epitype (designated here)*. BRAZIL. Rio Grande do Sul, Porto Alegre, Campus Vale UFRGS, 30°01′59″S 51°13′48″W, 3 m, 1 Apr. 2014, leg. Fazolino E. P., 164 (ICN 177266, ITS nrDNA, LSU nrDNA, *ATP*6, *RPB*2 and *TEF*-1α GenBank sequences: MG817719, MG817730, MH061923, MH061928, MH061937, respectively).

*Description*. Expanded basidiomata 85–130 mm length × 25–50 mm wide. Peridium composed of three layers (Fig 9A). Exoperidium white to light brown ($N_{00}Y_{00}M_{00}$–$N_{40}Y_{70}M_{30}$) smooth base, cracked into gray to dark brown ($N_{70}Y_{00}M_{00}$–$N_{90}Y_{80}M_{40}$) scales above. Single branched, white rhizomorph ($N_{00}M_{00}C_{00}$). Receptacle 3 to 4 arms united above, free in the base, whitish yellow to light orange ($Y_{70}M_{10}C_{00}$–$Y_{99}M_{40}C_{20}$), arms even thickness throughout its length (Fig 9B–9E, arrows in B and C indicating glebifers), outer face with a marked longitudinal groove (Fig 9F, see the arrow). Membranous glebifers adhered to the anterolateral angles of the arms, triangular, quadrangular or irregular, regularly spaced along the full length of the arms (Fig 9G–9I, see the arrows), glebifers covered with glebal mass.

Apical exoperidium composed of filamentous and elongated hyphae (Fig 9J and 9K). Filamentous hyphae, branched, regularly septate 2.5–9.2 μm diam., hyaline, straight and regular walls, 0.2–1.0 μm thick. Elongated hyphae look like the filamentous hyphae separated in the septa, 6.5–24.0 × 4.0–12.2 μm, brown lumen, straight and regular walls, 0.3–1.3 μm thick. Rhizomorphs composed of filamentous hyphae, branched, irregularly septate, 1.5–7.5 μm diam., hyaline, straight and regular walls, 0.3–1.0 μm thick (Fig 9L). Basidiospores cylindrical, 3.0–4.7 × 1.1–2.3 ($x = 3.7 \pm 0.2 \times 1.9 \pm 0.2$, Qm = 2.22 ± 0.15) μm, smooth, with an inner gutule at each end of the length, hyaline in KOH (Fig 9M) and Congo red, inamyloid and cyanophilous.

*Habit and habitat*. Epigeous and solitary, lignicolous in semi-deciduous forest, on decaying wood and on litter.

*Known distribution*. Neotropic: Brazil, Santa Catarina and Rio Grande do Sul [3,10,45].

*Additional material examined*. BRAZIL. Santa Catarina, Blumenau, leg. Möller (B700016470, photograph); *ibid*., leg. Möller 39, (HBG 024640, in alcohol); *ibid*. 15 Aug. 1891, leg. Möller (**sintype, designated here**: photograph reproduced in Möller [3]: Table III, Fig 1A); *ibid*. 02 Feb. 1893, leg. Möller (**sintype, designated here**: B700016469!, photograph reproduced in Möller [3]: Table III, Fig 3); *ibid* 07 Feb. 1893, leg. Möller (**sintype, designated here**: photograph reproduced in Möller [3]: Table III, Fig 2). Rio Grande do Sul, São Leopoldo, 1906, leg. Rick (PACA 12550); *ibid*. 1907, leg. Rick (PACA 12552); *ibid*. 1932, leg. Rick (PACA 12551). Rio Grande do Sul, Porto Alegre, Morro Santana, 30°01′59″S 51°13′48″W, 17 May 2011, leg. L. Trierveiler-Pereira 230 (ICN 176968, ITS nrDNA, LSU nrDNA, *ATP*6 and *TEF*-1α GenBank sequences: MG817718, MG817729, MH061922, MH061936, respectively). Rio Grande do Sul, Porto Alegre, Campus Vale UFRGS, 30°01′59″S 51°13′48″W, 3 m, 02 Apr. 2014, leg. Fazolino E. P., 165 (ICN 177267).

*Comments. Blumenavia rhacodes*, the type of the genus, presents thick arms of the same width, delimited grooves in the outer faces of the arms and membranous glebifers regularly spaced. An authentic Möller collection (B700016478!) from Blumenau (Fig 9E) has been designated as lectotype in this paper. However, since it was not possible to obtain molecular data from this collection; and these data are necessary to clearly separate *B. rhacodes* from other closely related species, such as *B. baturitensis* sp. nov, *B. heroica* sp. nov. and *B. toribiotalpaensis*, an epitype is also here designated (Fig 9B).

The phylogenetic trees (Fig 1) reveal *B. rhacodes* as a sister clade to *B. baturitensis* sp. nov. *Blumenavia rhacodes* is morphologically similar to *B. baturitensis* sp. nov, *B. heroica* sp. nov. and *B. toribiotalpaensis* by the arm color; but, *B. toribiotalpaensis* and *B. baturitensis* sp. nov. have irregularly shaped glebifers along the arms, only filamentous hyphae in the apical exoperidium and the arms are thinned from the middle to the top of the basidiomata. On the other hand, *B. heroica* sp. nov. has apical exoperidium composed of filamentous and subglobose to ellipsoid hyphae and the basidiomata up to 80mm length, mine while, *B. rhacodes* has filamentous and elongated hyphae in the apical exoperidium and the basidiomata above to 80mm length.

**Blumenavia toribiotalpaensis Vargas-Rodriguez, Mycotaxon 94: 8 (2005), Fig 10.** *Holotype*. MEXICO. Jalisco, Talpa de Allende, "Ojo de Agua del Cuervo oeste ao cumbre de los Arrastrados", 20°11″N 105°16″W, 1800 m, 10 Sept. 2002, leg. Yalma L. Vargas-Rodríguez 240, Javier Currel, António Vázquez, Toribio Quintero (BPI 870955!, ITS nrDNA, LSU nrDNA, *RPB*2 and *TEF*-1α GenBank sequences: MG817724, MG817732, MH061932, MH061941, respectively).

*Description*. Expanded basidiomata 121–153 mm length × 38–56 mm wide. Peridium composed of three layers. Exoperidium light yellow to dark yellow ($N10C00Y30–N_{10}C_{00}Y_{60}$) smooth base, cracked into gray ($N_{70}C_{00}Y_{20}$) from the middle until dark gray ($N_{90}C_{00}Y_{10}$) in the apical part. Single branched, white rhizomorph ($N_{00}M_{00}C_{00}$). Receptacle 3 to 4 arms united above, free in the base, whitish yellow to light orange ($Y_{70}M_{10}C_{00}–Y_{99}M_{40}C_{10}$), the arms are thinned from the middle to the top of the basidiomata, outer face with a marked longitudinal groove (Fig 10A–10E). Membranous glebifers adhered to the anterolateral angles of the arms, formed by large lacerated projections, slightly spaced from one another at apex to the upper half (Fig 10F), with small, more spaced, torn and twisted points to the base (Fig 10G and 10H), glebifers covered with glebal mass.

Apical exoperidium composed of filamentous, branched hyphae, irregularly septate, 2.1–7.1 μm diam, hyaline, straight and regular walls, 0.2–0.8(–1.1) μm thick (Fig 10I and 10J). Rhizomorphs composed of filamentous hyphae, branched, irregularly septate, 1.2–6.1 (–8.0) μm diam., hyaline, straight and regular walls, some of them slightly winding, 0.1–0.9 μm thick

(Fig 10K). Basidiospores cylindrical, 3.3–4.9 × 1.4–2.6 ($x$ = 4.0 ± 0.2 × 1.8 ± 0.2, Qm = 2.29 ± 0.18, smooth, with an inner gutule at each end of the length, hyaline in KOH (Fig 10L) and Congo red, inamyloid and cyanophilous.

*Habit and habitat*. Epigeous and solitary, lignicolous in mesophilic mountain forest. (*Acer* L., *Abies* Mill., *Podocarpus* L'Her ex Pers.) on decaying wood of *Pinus* L.

*Known distribution*. Neotropic: Mexico, Jalisco [7].

*Additional material examined*. MEXICO. Jalisco, Talpa de Allende, "Ojo de Agua del Cuervo oeste ao cumbre de los Arrastrados", 20°11″N 105°16″W, 1800 m, 10 Sept. 2002, leg. Y. L. Vargas-Rodríguez 240a, J. Currel, A. Vázquez-García, T. Quintero (**isotype**: IBUG 240a!); *ibid*. 13 Sept. 2005, leg. Y. L. Vargas-Rodríguez 454, A. Vázquez-García (**paratype**: IBUG 454); *ibid*. 13 Sept. 2005, leg. Y. L. Vargas-Rodríguez 456, A. Vázquez-García (**paratype**: IBUG 456, ITS nrDNA, *RPB*2 and *TEF*-1α GenBank sequences: MG817723, MH061931, MH061940, respectively); *ibid*. 13 Sept. 2005, leg. Y. L. Vargas-Rodríguez 462, A. Vázquez-García (**paratype**: IBUG 462).

*Comments*. *Blumenavia toribiotalpaensis* was synonymized by Calonge et al. [12] with *B. rhacodes* for sharing distributed glebifers throughout the receptacle. However, as confirmed in our study *B. toribiotalpaensis* glebifers do not maintain the constant form that occurs in *B. rhacodes*. Moreover, *B. rhacodes* has arms of uniform dimensions along the entire length, and in *B. toribiotalpaensis* arms become thinner from the middle to the top of the basidiomata. *Blumenavia baturitensis* sp. nov. resembles *B. toribiotalpaensis* for whitish yellow arms thinning from the middle to the top of basidiomata and a filamentous exoperidium. However, *B. baturitensis* sp. nov. has tenticular glebifers and the grooves in the external face of the arms are absent. The molecular analyses confirm *B. toribiotalpaensis* is phylogenetically separate from species with yellowish basidiomata, since it is the sister species of *B. crucis-hellenicae* sp. nov., a species with white basidiomata.

**Blumenavia usambarensis Hennings, Bot. Jb. 33: 37 (1902) [1904], Fig 11**. *Holotype*. TANZANIA. Usambara Montains, Kummer 60, 19 Jan. 1900 (B 700016468!).

*Description*. Expanded basidiomata 80 mm length × 25–30 mm wide. Peridium composed of three layers. Exoperidium whitish to light brown (N00Y30M00–N30Y60M30) smooth. Single, branched, white rhizomorph (N00M00C00). Receptacle 3 to 5 arms united above, free in the base, white (N00M00C00), arms of even thickness throughout its length, without grooves (Fig 11A and 11B). Membranous glebifers adhered to the anterolateral angles of the arms, triangular, quadrangular or irregular, regularly spaced along the upper part of the arms (Fig 11C), glebifers covered with glebal mass.

Apical exoperidium composed of filamentous, branched hyphae, regularly septate, 2.1–7.8 μm diam., hyaline, straight and regular walls, 0.3–0.5 μm thick. Rhizomorphs composed of filamentous hyphae, branched, irregularly septate, 5.7–16.1 μm diam., hyaline, straight and regular walls, slightly winding, 1.0–4.7 μm thick (Fig 11D). Basidiospores cylindrical, 2.6–3.3 × 1.1–1.5 ($x$ = 3.1 ± 0.2 × 1.3 ± 0.1, Qm = 2.41 ± 0.20) μm, smooth, with an inner gutule at each end of the length, hyaline in KOH (Fig 11E) and Congo red, inamyloid and cyanophilous.

*Habit and habitat*. Epigeous and solitary, saprotrophic in mountain vegetation.

*Known distribution*. Afrotropic: Tanzania [6].

*Comments*. *Blumenavia usambarensis* described by Hennings [6] was synonymized with *B. angolensis* by Dring [4] due to the white arms and glebifers distributed only in the quarter top of basidiomata. However, *B. usambarensis* has cylindrical and smaller basidiospores (2.6–3.3 × 1.1–1.5) in relation to *B. angolensis* (3.3–4.0 × 1.4–1.8). Another character of *B. usambarensis* is the presence of thick-walled rhizomorph hyphae (1.0–4.7 μm), which distinguishes it from other species of *Blumenavia* (0.1–1.1 μm).

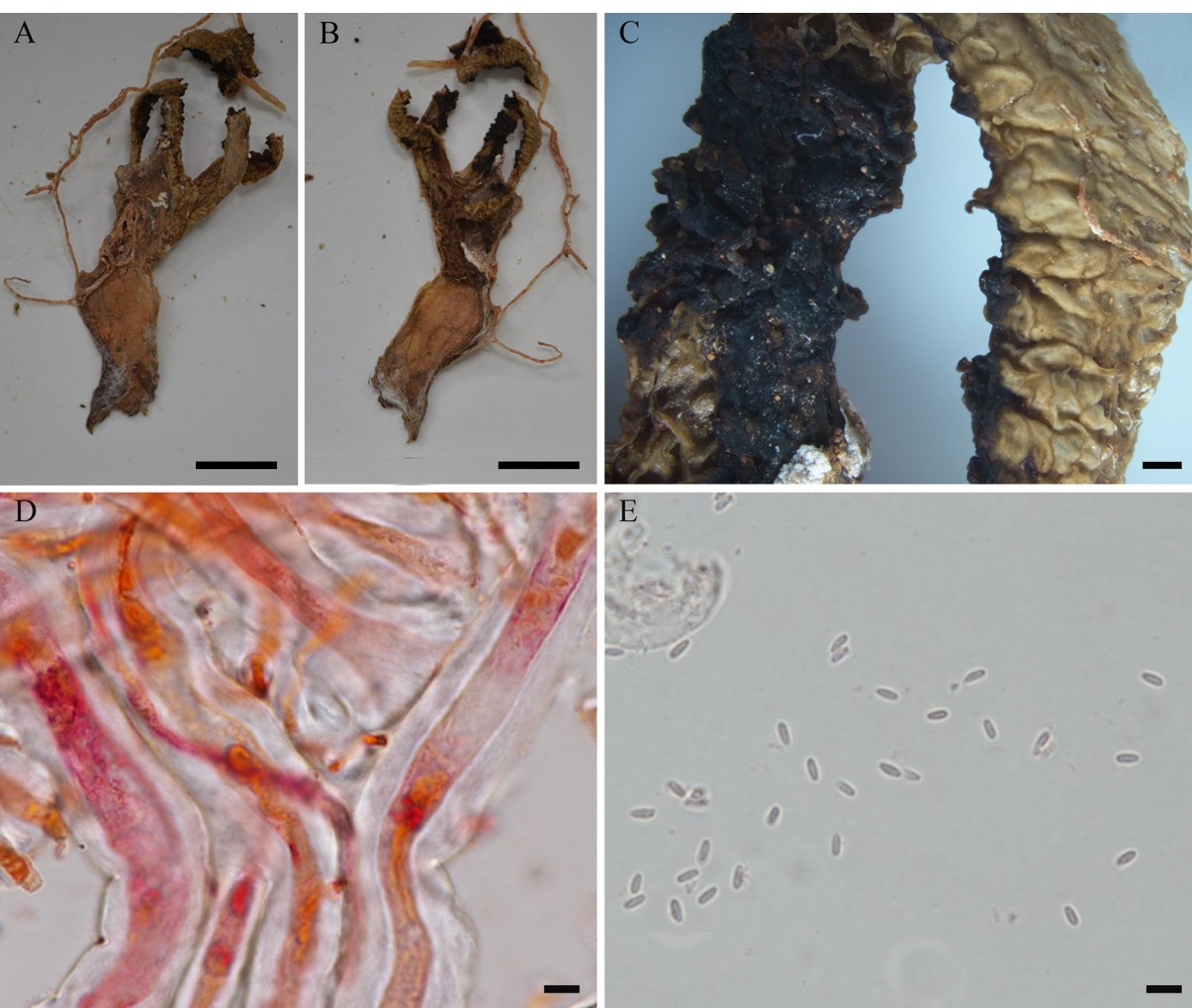

**Fig 11. *Blumenavia usambarensis*, B 700016468, holotype.** (A, B) Dry basidiomata. (C) Arm edge and glebifer. (D) Rhizomorph hyphae in 1% Congo red. (E) Basidiospores in 5% KOH. (A, B) bar = 10 mm; (C) bar = 1 mm; (D–E) = 5 μm.

## Conclusion

Scarce molecular data, together with the use of a limited set of informationally poor morphological characters, led to misidentifications and synonymization in *Blumenavia* species. There are a small number of macromorphological characters that are useful for delimiting species, but some of these must be observed while the basidiomata are still fresh, such as color oh the basidiomata as well as the arrangement and morphology of the glebifers, since they become difficult to observe in fungarium collections. Even so, the analysis of morphological characters, integrated with molecular data, as well as geographical distribution allowed the definition of diagnostic characters for *Blumenavia* species, affirming the hypothesis of this work. The results obtained reaffirm the importance of the integration with different data, and the importance of comparative revision work with materials deposited in collections, especially type species, to delimit and identify the species in *Blumenavia*, as in another gasteroid with confusing taxonomy.

## Supporting information

**S1 Table. Microscopic uninformative characters in *Blumenavia*.** Basal and apical arms, glebifers, basal exoperidia, mesoperidia and endoperidia.
(XLSX)

## Acknowledgments

ACMR, GCSM, TA and TSC thank the funding agency CAPES, and RJF to CNPq, for the scholarships. Acknowledgments are also due to the herbarium curators (B, BPI, HBG, IBUG, ICN, XAL, MBM, PACA, SMDB) by the loan of exsiccates, and for sending photographs of exsiccates; and the collectors for sending pictures of some specimens. Thanks to Association des amis des cryptogames (A.D.A.C.), Claire Margerie (Administration for Publications Scientifiques, Muséum national d´Histoire natural), and Jérôme Degreef for permission to reproduce the image of *B. angolensis* from Degreef et al. [8]; to Gina Fullerlove (Head of Publishing of Royal Botanic Gardens, Kew) for permission to reproduce the image of *B. angolensis* from Dring [4]; to Dennis Desjardin for permission to reproduce the image of *B. angolensis* from Desjardin and Perry [11]; to Yalma L. Vargas- Rodríguez and J. Antonio Vázquez-García for permission to reproduce the images of *B. toribiotalpaensis* from Vargas-Rodríguez and Vázquez-García [7]. The authors give sincere thanks to Marian Glenn for English revision.

## Author Contributions

**Conceptualization:** Gislaine C. S. Melanda, Thiago Accioly, María P. Martín, Iuri G. Baseia.

**Data curation:** Gislaine C. S. Melanda, Thiago Accioly, Renato J. Ferreira, María P. Martín, Iuri G. Baseia.

**Formal analysis:** Gislaine C. S. Melanda, Thiago Accioly, Renato J. Ferreira, Ana C. M. Rodrigues, Tiara S. Cabral, Gilberto Coelho, Tine Grebenc, María P. Martín, Iuri G. Baseia.

**Funding acquisition:** Marcelo A. Sulzbacher, Tine Grebenc, María P. Martín, Iuri G. Baseia.

**Investigation:** Gislaine C. S. Melanda, Thiago Accioly, Renato J. Ferreira, María P. Martín, Iuri G. Baseia.

**Methodology:** Gislaine C. S. Melanda, Thiago Accioly, Renato J. Ferreira, Ana C. M. Rodrigues, Tiara S. Cabral, Gilberto Coelho, Marcelo A. Sulzbacher, Vagner G. Cortez, Tine Grebenc, María P. Martín, Iuri G. Baseia.

**Project administration:** Iuri G. Baseia.

**Resources:** Marcelo A. Sulzbacher, Tine Grebenc, María P. Martín, Iuri G. Baseia.

**Supervision:** María P. Martín, Iuri G. Baseia.

**Validation:** Gislaine C. S. Melanda, María P. Martín, Iuri G. Baseia.

**Visualization:** Gislaine C. S. Melanda, Thiago Accioly, María P. Martín, Iuri G. Baseia.

**Writing – original draft:** Gislaine C. S. Melanda, Thiago Accioly, Renato J. Ferreira, Ana C. M. Rodrigues, Tiara S. Cabral, Gilberto Coelho, Marcelo A. Sulzbacher, Tine Grebenc, María P. Martín, Iuri G. Baseia.

**Writing – review & editing:** Gislaine C. S. Melanda, Thiago Accioly, María P. Martín, Iuri G. Baseia.

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
