## [Decision Letter · Decision Letter 0]

26 Feb 2020

PONE-D-20-01494

Diversity trapped in cages: revision of Blumenavia Möller (Clathraceae, Basidiomycota) reveals three hidden species

PLOS ONE

Dear Mrs. Melanda,

Thank you for submitting your manuscript to PLOS ONE. After careful consideration, we feel that it has merit but does not fully meet PLOS ONE’s publication criteria as it currently stands. Therefore, we invite you to submit a revised version of the manuscript that addresses the points raised during the review process.

===

All three reviewers found your study interesting. However, they also suggested revisions. Although overall it is a minor revision, they are many places that the authors have to pay close attention. Especially, the authors need to  respond carefully to the reviewers 1 and 3 as they requested a thorough edition. 

We would appreciate receiving your revised manuscript by Apr 11 2020 11:59PM. To enhance the reproducibility of your results, we recommend that if applicable you deposit your laboratory protocols in protocols.io, where a protocol can be assigned its own identifier (DOI) such that it can be cited independently in the future. For instructions see: http://journals.plos.org/plosone/s/submission-guidelines#loc-laboratory-protocols

We look forward to receiving your revised manuscript.

Kind regards,

Soo Chan Lee

Academic Editor

PLOS ONE

Journal Requirements:

2. In your Methods section, please provide additional details regarding the samples used in your study and ensure you have described the source. For more information regarding PLOS' policy on materials sharing and reporting, see https://journals.plos.org/plosone/s/materials-and-software-sharing#loc-sharing-materials.

3. Please take this opportunity to be sure you have met all of our guidelines for new species. When publishing papers that describe a new fungal taxon name, PLOS aims to comply with the requirements of the International Code of Nomenclature for algae, fungi, and plants (ICN). The following guidelines for publication in an online-only journal have been agreed such that any scientific fungal name published by us is considered effectively published under the rules of the Code. Please note that these guidelines differ from those for zoological nomenclature.

Effective January 2012, "the description or diagnosis required for valid publication of the name of a new taxon" can be in either Latin or English. This does not affect the requirements for scientific names, which are still to be Latin.

Also effective January 2012, the electronic PDF represents a published work according to the ICN for algae, fungi, and plants. Therefore the new names contained in the electronic publication of a PLOS ONE article are effectively published under that Code from the electronic edition alone, so there is no longer any need to provide printed copies.

For proper registration of the new taxon, we require two specific statements to be included in your manuscript.

a)    In the Results section, the globally unique identifier (GUID), currently in the form of a Life Science Identifier (LSID), should be listed under the new species name, for example:

Hymenogaster huthii. Stielow et al. 2010, sp. nov. [urn:lsid:indexfungorum.org:names:518624]

You will need to contact either Mycobank or Index Fungorum to obtain the GUID (LSID).

b)    In the Methods section, include a sub-section called "Nomenclature" using the following wording (this example is for taxon names submitted to MycoBank; please substitute appropriately if you have submitted to Index Fungorum and use the prefix http://www.indexfungorum.org/Names/NamesRecord.asp?RecordID= ):

The electronic version of this article in Portable Document Format (PDF) in a work with an ISSN or ISBN will represent a published work according to the International Code of Nomenclature for algae, fungi, and plants, and hence the new names contained in the electronic publication of a PLOS ONE article are effectively published under that Code from the electronic edition alone, so there is no longer any need to provide printed copies.

In addition, new names contained in this work have been submitted to MycoBank from where they will be made available to the Global Names Index. The unique MycoBank number can be resolved and the associated information viewed through any standard web browser by appending the MycoBank number contained in this publication to the prefix http://www.mycobank.org/MB/. The online version of this work is archived and available from the following digital repositories: [INSERT NAMES OF DIGITAL REPOSITORIES WHERE ACCEPTED MANUSCRIPT WILL BE SUBMITTED (PubMed Central, LOCKSS etc)].

All PLOS ONE articles are deposited in PubMed Central and LOCKSS. If your institute, or those of your co-authors, has its own repository, we recommend that you also deposit the published online article there and include the name in your article.

A complete explanation of our guidelines for publishing new species can be found on our website: http://www.plosone.org/static/guidelines#fungal

Reviewers' comments:

Reviewer's Responses to Questions

**Comments to the Author**

1. Is the manuscript technically sound, and do the data support the conclusions?

Reviewer #1: Yes

Reviewer #2: Yes

Reviewer #3: Yes

2. Has the statistical analysis been performed appropriately and rigorously? 

Reviewer #1: Yes

Reviewer #2: Yes

Reviewer #3: Yes

3. Have the authors made all data underlying the findings in their manuscript fully available?

Reviewer #1: Yes

Reviewer #2: Yes

Reviewer #3: Yes

4. Is the manuscript presented in an intelligible fashion and written in standard English?

Reviewer #1: Yes

Reviewer #2: Yes

Reviewer #3: Yes

5. Review Comments to the Author

Reviewer #1: The authors have done a very thorough revision of Blumenavia genus, including type specimens. This is a poorly studied group with rare species mainly because of their receptacles ephemeral and fragile. The manuscript presents original data and relevant information. Some modification are necessary to in order to render it accaptable for publication. My suggestions are included in the attached PDF document.

Reviewer #2: The authors present a highly interesting topic for the criptic taxa of the genus Blumenavia, combining morphological and molecular data of the analyzed species, their ecology, as well as comparative revision of the material form collections, especially type species.

The presented data are sufficient, representative and well documented, presenting new and original information. The interpretations are logical and well supported.

The text is concise and clearly explains topics, methodologies, data and conclusions.

The title of the paper clearly characterizes its content.

The abstract is clear and informative and comprise adequately the content of the manuscript.

The applied methods are appropriate, presented clearly and understandable.

The descriptions of species are fairly well performed and relevant.

Figures and tables are clearly presented and correctly labeled. They are instructive and captions unambiguously explain pictured details. Important features from all available material are presented with clear photos. Additionally, including the reprints of the old collections, representing all data at the same place, is a very good way for comparison.

All the references are adequate and necessary, considering recent research.

Conclusions are highly supported by the presented data and the results.

Minor issues:

p. 9 Molecular phylogeny

The number of the specimens and the number of the sequences written in the text is not the same as shown in Table 1. Please check it.

p. 14, 322 B. heroica should be changed with B. crucis-hellenicae. (B. heroica belongs to the white colored species and B. crucis-hellenicae is with yellowish color, as it is shown in all other parts for both species – key, description, photos).

I recommend this paper as acceptable for publication.

Reviewer #3: Dear Authors,

Thank you for the opportunity to review a nice taxonomic manuscript. It would be more traditional to see it in Mycologia or Taxon magazines, but regarding novel adds to the (still) not quite traditional methods to delimit species boundaries like phylogeny PLoS definitely can serve as a proper place for it. I feel confident to recommend your manuscript to be published with minor revisions. Also, I have a few questions, below.

1. I believe Table 1 better belongs to the Supplements.

Material and Methods

Phylogenetic Analysis

2. Did you obtained all sequences of the specimens from genus Blumenavia you have used for your reconstructions? Have you compared your sequences to the available ones in GenBanks and how you find their identity/ similarity?

Results and Discussion

Molecular phylogeny, lines 196-198

3. For this study, DNA sequences of 12 specimens of Blumenavia were obtained: 9 ITS, 8 LSU, 6 ATP6, 9 RPB2 and 9 TEF-1a (Table 1). Sequences from B. usambarensis were not obtained.

better to modify

For this study, DNA sequences of 12 specimens of Blumenavia were obtained: 9 ITS, 8 LSU, 6 ATP6, 9 RPB2 and 9 TEF-1a (Table 1),except B. usambarensis.

4. Did all trees (MP, ML and BI) have the same topology?

Blumenavia angolensis (Welw. & Curr.) Dring, Kew Bull. 35(1): 53 (1980), description

5. I believe, "covered with glebal mass." in line 279 has to be removed.

6. Cannot access the alignments at http://purl.org/phylo/treebase/phylows/study/TB2:S25096?x-access192 code=cb0af291cea482c1bfa56649485eb9b2&format=html

I wish you good luck with this publication and hope to see it soon in PLoS.

6. PLOS authors have the option to publish the peer review history of their article (what does this mean?). If published, this will include your full peer review and any attached files.

Reviewer #1: No

Reviewer #2: No

Reviewer #3: No

---

## [Author Response · Author response to Decision Letter 0]

3 Apr 2020

Dear editor, thank you very much for giving us the opportunity to send a revised manuscript. Thank you also to Reviewers for some of his/her constructive comments. 

We’ve checked all guidelines and verified the correction of any mistake. 

The response to each comment is on file Response to Reviewers attached in the review submission.

---

## [Decision Letter · Decision Letter 1]

16 Apr 2020

Diversity trapped in cages: revision of Blumenavia Möller (Clathraceae, Basidiomycota) reveals three hidden species

PONE-D-20-01494R1

Dear Dr. Melanda,

We are pleased to inform you that your manuscript has been judged scientifically suitable for publication and will be formally accepted for publication once it complies with all outstanding technical requirements.

With kind regards,

Soo Chan Lee

Academic Editor

PLOS ONE

Additional Editor Comments (optional):

Reviewers' comments:

Reviewer's Responses to Questions

**Comments to the Author**

1. If the authors have adequately addressed your comments raised in a previous round of review and you feel that this manuscript is now acceptable for publication, you may indicate that here to bypass the “Comments to the Author” section, enter your conflict of interest statement in the “Confidential to Editor” section, and submit your "Accept" recommendation.

Reviewer #1: (No Response)

Reviewer #2: All comments have been addressed

Reviewer #3: All comments have been addressed

2. Is the manuscript technically sound, and do the data support the conclusions?

Reviewer #1: Yes

Reviewer #2: Yes

Reviewer #3: Yes

3. Has the statistical analysis been performed appropriately and rigorously? 

Reviewer #1: Yes

Reviewer #2: Yes

Reviewer #3: Yes

4. Have the authors made all data underlying the findings in their manuscript fully available?

Reviewer #1: Yes

Reviewer #2: Yes

Reviewer #3: Yes

5. Is the manuscript presented in an intelligible fashion and written in standard English?

Reviewer #1: Yes

Reviewer #2: Yes

Reviewer #3: Yes

6. Review Comments to the Author

Reviewer #1: Dear Authors,

Congratulations for the great work.

The manuscript is much improved, with modified text to better understanding and adding relevant informations on the study of the genus Blumenavia.

Most of the suggestions were accepted and others were justified.

Very nice work!

Reviewer #2: The authors have adequately addressed the comments.

The manuscript sound technically and the data support the conclusions.

The statistical analysis has been performed appropriately and rigorously.

The authors have made all data fully available in their manuscript.

The manuscript is presented in an intelligible fashion and written in standard English.

I recommend this manuscript for publication.

Reviewer #3: Dear Authors,

Thank you for your thorough revision through the manuscript and addressing all my concerns and comments. I believe, your manuscript is good to go and can now be published with no further reservations.

Sincerely,

Reviewer.

7. PLOS authors have the option to publish the peer review history of their article (what does this mean?). If published, this will include your full peer review and any attached files.

Reviewer #1: No

Reviewer #2: No

Reviewer #3: Yes: Andrii P. Gryganskyi

---

## [Editor Report · Acceptance letter]

21 Apr 2020

PONE-D-20-01494R1 

Diversity trapped in cages: revision of *Blumenavia* Möller (Clathraceae, Basidiomycota) reveals three hidden species 

Dear Dr. Melanda:

I am pleased to inform you that your manuscript has been deemed suitable for publication in PLOS ONE. Congratulations! Your manuscript is now with our production department. 

With kind regards,

on behalf of

Dr. Soo Chan Lee 

Academic Editor

PLOS ONE